# Anthropogenic air pollutants strongly interact with natural aerosols over the eastern China seas: key processes, size distributions, and seasonalities

Shengqian Zhou<sup>1</sup>¶, Zongjun Xu<sup>1</sup>¶, Ying Chen<sup>1,2,3</sup>, Mingtao Zhao<sup>1</sup>, Yifei Li<sup>1</sup>, Ke Yan<sup>1</sup>

<sup>1</sup>Shanghai Key Laboratory of Atmospheric Particle Pollution Prevention, Department of Environmental Science & Engineering, Fudan University, Shanghai 200438, China

<sup>2</sup>Institute of Eco-Chongming (IEC), National Observations and Research Station for Wetland Ecosystems of the Yangtze Estuary, Shanghai 200062, China

<sup>3</sup>Institute of Atmospheric Sciences, Fudan University, Shanghai 200438, China

These authors contributed equally to this work.

Correspondence to: Ying Chen (yingchen@fudan.edu.cn)

Abstract. Marine aerosols play important roles in climate, marine biogeochemistry, and coastal air quality. Over the eastern China seas adjacent to densely populated East Asia, aerosols are mutually affected by anthropogenic pollution and natural emissions. However, the impacts of anthropogenic-natural interactions on aerosol composition and properties are not well constrained due to limited systematic observations. Here we characterized the composition of size-resolved aerosols over this region across four seasonal cruise campaigns, identifying major aerosol sources and influencing processes. Aerosol mass concentrations typically show trimodal distributions, with fine-mode mass dominating in spring and winter due to strong influences of continental pollution. However, in 50.9% of samples, continental secondary aerosols are highly aged, lacking a fine-mode NO<sub>3</sub><sup>-</sup> peak. Gaseous HNO<sub>3</sub> evaporated from continental secondary aerosols and anthropogenic NO<sub>x</sub> react with natural dust and sea spray aerosols (SSA), forming coarse-mode NO<sub>3</sub><sup>-</sup>, which contributes 43.2% and 12.7% of total NO<sub>3</sub><sup>-</sup>, respectively. This shifts NO<sub>3</sub><sup>-</sup> from fine to coarse mode, altering the spatial pattern of nitrogen deposition and its ecological effects. Additionally, 27.7% of SSA Cl<sup>-</sup> is depleted on average, reaching 40.8% in summer, which is an important source of reactive halogens that affect ozone chemistry. Shipping emissions contribute to more than 20% of SO<sub>4</sub><sup>2-</sup> in spring and summer before the International Maritime Organization's 2020 regulation, but this contribution likely decreases by one order of magnitude thereafter. This analysis highlights the importance of anthropogenic-natural interactions over coastal seas, underscoring the need for further studies to assess their subsequent environmental impacts.

### 1 Introduction

Atmospheric aerosol, originating from diverse natural and anthropogenic sources such as sea spray, dust, biomass burning, industrial processing, transportation, and residential emissions (Textor et al., 2006; Soulie et al., 2024), play a critical role in Earth system (Song et al., 2017a; Song et al., 2017b; Pöschl, 2005; Mahowald, 2011; Duce et al., 2008). Aerosols influence the atmospheric energy budget by directly scattering and absorbing shortwave radiation and indirectly altering cloud

reflectivity, coverage, and lifetime (Seinfeld and Pandis, 2016; Lohmann and Feichter, 2005). Through long-range transport and subsequent deposition, aerosols deliver nutrients such as nitrogen, phosphorus, and iron to distant regions, serving as an essential pathway for nutrient input and playing a significant role in both marine and terrestrial ecosystems (Mahowald, 2011). Additionally, aerosols are a major component of atmospheric pollution (e.g., haze), which poses significant health risks to human populations (Pöschl, 2005; Song et al., 2017a). The climatic, ecological, and health impacts of aerosols are closely tied to both their chemical composition and size distribution, which are controlled by emission sources and atmospheric processes. To fully understand the environmental effects of aerosols in a specific region, it is crucial to characterize their properties, identify their predominant sources, and investigate how they evolve in the atmosphere.

Eastern China is one of the most densely populated regions in the world, where intense human activities release large amounts of gaseous precursors and primary aerosols into the atmosphere, leading to severe haze issues (Wang et al., 2015; An et al., 2019; Wang et al., 2013). It was estimated that China's anthropogenic emissions of SO<sub>2</sub>, NO<sub>3</sub>, NH<sub>3</sub>, and PM<sub>2.5</sub> in 2015 were 18.4, 24.3, 14.1, and 11.3 Tg, respectively—accounting for nearly half of Asia's total emissions (Kurokawa and Ohara, 2020). Additionally, East Asia serves as a major global dust source, contributing ~11% of global atmospheric dust 45 budget (Kok et al., 2021). Driven by East Asian monsoon, these anthropogenic pollutants and dust aerosols are transported across the North Pacific Ocean, where they undergo complex mixing and chemical interactions with marine aerosols, significantly modifying their physicochemical properties (Hsu et al., 2010; Moldanová and Ljungström, 2001). Concurrently, dry and wet depositions of the aerosols can influence marine phytoplankton and ecosystems (Wang et al., 2017; Meng et al., 2016). For example, the deposition of anthropogenic aerosols has been the primary driver of increasing excess nitrogen (relative to phosphorus) concentrations in the northwest Pacific and its marginal seas since 1980, potentially enhancing primary productivity in nitrogen-limited areas (Kim et al., 2011). Notably, the marginal seas adjacent to eastern China (i.e., the East China Sea, the Yellow Sea, and the Bohai Sea) represent the first geographical node along this long-range transport pathway. Therefore, both atmospheric composition and oceanic ecosystem in these regions are strongly affected by continental outflow. In addition, eastern Chinese coast hosts numerous world-class ports, where intensive shipping activities emit substantial aerosols and gaseous precursors, further complicating aerosol-marine interactions (Feng et al., 2019).

In such a polluted marine atmospheric environment over the marginal seas of eastern China, aerosol concentrations and properties are shaped by multiple factors and processes, including anthropogenic and natural emissions, air mass transport, and chemical reactions. This region provides a unique setting to study the complex interactions between anthropogenic pollutants and natural aerosols and to assess their climatic and ecological impacts. Investigating aerosol chemical composition, size distribution, and sources is crucial for improving our understanding of these interactions and enhancing the validation of chemical transport models.

60

To characterize the aerosols in marine boundary layer, three main approaches have been utilized: fixed-point island observation, cruise/flight measurements, and satellite remote sensing. Several long-term island observational stations have been established, such as Huaniao Island off the Yangtze River Delta (Wang et al., 2016; Zhou et al., 2021), Cape Hedo in

Okinawa (Kunwar and Kawamura, 2014), and Gosan in Jeju Island (Wang et al., 2009). While island-based observations provide long-term continuous data, their spatial representativeness is relatively low. Cruise and flight measurements, on the other hand, cover larger spatial areas, particularly in remote oceanic regions where ground-based observations are impractical. However, they are typically conducted during short-term field campaigns, limiting their ability to capture seasonal variations. To date, a systematic study of size-dependent aerosol composition across all four seasons based on cruise or flight measurements remains scarce. Although satellite remote sensing offers long-term, large-scale observations, it is limited in retrieving detailed aerosol chemical composition.

In this study, we collected size-segregated aerosol samples during four cruises in different seasons over the Yellow Sea and the East China seas (YECS) and measured the concentrations of major aerosol components. Based on the measurements, we investigated the chemical signatures, dominant sources, and key atmospheric processes influencing aerosols in the region. Furthermore, we examined how and to what extent human activities modify aerosol properties in this polluted marine atmosphere, along with their seasonal variations. Our results enhance understanding of aerosol characteristics and their environmental impacts in this transitional region from heavily polluted continents to relatively pristine oceans.

## 2 Methodology

## 2.1 Aerosol sampling

Four research cruises were conducted across the YECS during the spring of 2017 (27 March – 15 April, R/V *Dong Fang Hong II*), the summer of 2018 (26 June – 19 July, R/V *Dong Fang Hong II*), the autumn of 2020 (10 October – 29 October, R/V *Xiang Yang Hong XVIII*), and the winter of 2019–2020 (26 December 2019 – 18 January 2020, R/V *Dong Fang Hong III*). A total of 53 sets of size-segregated samples were collected—9 in spring, 10 in summer, 16 in autumn, and 18 in winter.

Sampling was conducted using an 11-stage Micro-Orifice Uniform Deposit Impactor (MOUDI, MSP Model 110-NR) at a flow rate of 30 L min<sup>-1</sup>. The impactor's 50% cutoff diameters are 18, 10, 5.6, 3.2, 1.8, 1.0, 0.56, 0.32, 0.18, 0.10, and 0.056 µm. Polytetrafluoroethylene (PTFE) membranes (Zeflour, PALL Life Sciences) were used in the first two cruises, while polycarbonate membranes (Millipore) were used in the latter two. Both membrane types have minimal background levels of water-soluble ions and metal elements, ensuring they did not interfere with the chemical measurements. To prevent contamination from the ship's exhaust, the MOUDI sampler was mounted at the front of the uppermost deck, and sampling was conducted only when the apparent wind came from the bow. Sampling durations generally ranged from 24 to 48 hours. Figure 1 illustrates the sailing tracks of all four cruises and the specific segments during the collection period of each sample.

The abovementioned samples were primarily used for analyzing water-soluble ions and trace elements. During the first two cruises, a second 11-stage MOUDI sampler equipped with quartz membranes (PALL Life Sciences) was also deployed simultaneously for organic measurements. Before sampling, quartz membranes were baked at 450 °C for 4 h to remove residual organic contaminants.

Figure 1. Tracks of four cruises the specific segments during the collection period of each MOUDI sample. Each number represents a sample.

### 2.2 Chemical analysis

Half of each PTFE or polycarbonate sample membrane was cut and ultrasonically extracted with 20 mL deionized water. Water-soluble ions (Na<sup>+</sup>, NH<sub>4</sub><sup>+</sup>, K<sup>+</sup>, Mg<sup>2+</sup>, Ca<sup>2+</sup>, Cl<sup>-</sup>, NO<sub>3</sub><sup>-</sup>, SO<sub>4</sub><sup>2-</sup>, and C<sub>2</sub>O<sub>4</sub><sup>2-</sup>) were analyzed by ion chromatography (DIONEX ICS-3000, Thermo) after filtration through a 0.45 μm PTFE syringe filter. For trace metal elements analysis, one-fourth of each sample membrane was digested with 7 mL of HNO<sub>3</sub> and 1mL of HF (both acids purified via a sub-boiling system) at 185°C for 30 min in a microwave digestion system (MARS5 Xpress, CEM). Trace elements, including Al, Ba,

Ca, Cd, Ce, Co, Cu, Eu, Fe, Ge, K, Mg, Mn, Mo, Na, Ni, Pb, Rb, Sb, Se, Sr, V, and Zn, were analyzed using inductively coupled plasma mass spectrometry (ICP-MS, PerkinElmer). Organic carbon (OC) and elemental carbon (EC) concentrations were determined using an OC/EC analyzer (DRI, 2001A), with a 0.544 cm<sup>2</sup> circular punch from quartz sample membrane directly analyzed by the instrument. However, due to the low EC concentrations and limited detectability (e.g., below the detection limit in 86% of samples during the summer campaign), the EC data were excluded from subsequent analyses.

Particulate matter (PM) mass concentrations across different size ranges were determined by weighing the sampling membranes (PTFE or polycarbonate) before and after collection using an analytical balance (Sartorius, readability: 0.01 mg).

# 2.3 Air mass backward trajectory

72-hour air mass backward trajectories arriving at the sampling sites (i.e., research vessel locations) during the four cruise campaigns were calculated using the Hybrid Single-Particle Lagrangian Integrated Trajectories (HYSPLIT) model (Stein et al., 2015). The calculations were driven by meteorological data from the Global Data Assimilation System (GDAS, 1° × 1°) (https://www.ncei.noaa.gov/products/weather-climate-models/global-data-assimilation). Backward trajectories were generated hourly, with a starting height of 100 m above sea level. For each trajectory, the time spent over ocean was calculated and then averaged over the sampling period corresponding to each MOUDI sample. Air temperature along each trajectory was extracted from the GDAS dataset based on the location, altitude, and timestamp of each trajectory point, and subsequently averaged for each sample.

## 2.4 Size-resolved positive matrix factorization

Size-resolved Positive Matrix Factorization (SR-PMF) was applied to analyze the main sources of aerosol components across different size ranges over the eastern China seas. Here the sources include both primary emission and secondary formation. Unlike conventional PMF approaches that use the time series of bulk aerosol chemical species as input (Wang et al., 2016; Dai et al., 2020; Crippa et al., 2013), our analysis incorporates both temporal variations and size distributions. Instead of summing concentrations across all size bins, we treated the chemical composition spectrum of each size range within a sample set as a separate instance for PMF input. Since each sample set contains 11 size bins (excluding  $D_p < 0.056$  µm due to the lack of element measurements), our dataset comprises 583 instances from 53 sample sets across four research cruises. The chemical species involved in SR-PMF analysis include nine water-soluble ions and 23 elements as mentioned in Section 2.2. The U.S. EPA's PMF 5.0 software was used for performing this analysis (Brown et al., 2015).

This SR-PMF approach offers several advantages over traditional methods. First, the number of instances is increased by one order of magnitude, which can mathematically enhance the robustness of the factorization results. Second, a certain chemical compound may originate from multiple different sources with similar chemical profiles and their contributions may also exhibit similar temporal variations due to dominant atmospheric transport patterns, which is difficult for traditional PMF to differentiate them. However, if the aerosols from these sources have distinct size distributions, incorporating size-resolved

data provides critical additional information, enabling PMF to differentiate and verify them more effectively. From another perspective, this method quantitatively reveals the typical size distribution of aerosols from each source.

Studies integrating size distribution with PMF models have been previously reported. For example, Guo et al. (2010) used PMF to separate different aerosol modes based on the water-soluble ion components in Beijing, shedding light on their formation mechanisms. However, this study did not combine other components such as trace elements. Beddows et al. (2015) combined the time series of PM<sub>10</sub> chemical composition with the temporal variations of particle number size distributions to identify aerosol sources in London, UK. In contrast, our study directly utilizes the chemical composition of size-segregated aerosols.

### 3 Results

### 145 3.1 PM mass concentration

The total PM mass concentration within the size range of  $0.056\text{-}18~\mu\text{m}$  was  $40.2\pm21.3~\mu\text{g}$  m<sup>-3</sup> (mean  $\pm$  1 standard deviation) in the spring of 2017,  $20.7\pm6.7~\mu\text{g}$  m<sup>-3</sup> in the summer of 2018,  $38.9\pm19.2~\mu\text{g}$  m<sup>-3</sup> in the autumn of 2020 and  $31.6\pm8.0~\mu\text{g}$  m<sup>-3</sup> in the winter of 2019-2020. These values were all more than twice lower than the total suspended particle (TSP) concentrations previously reported over the Huaniao island in coastal East China Sea (Guo et al., 2014). The aerosol mass concentration exhibited a trimodal distribution, with three distinct peaks at  $3.2\text{-}5.6~\mu\text{m}$  (coarse mode),  $0.56\text{-}1.0~\mu\text{m}$  (droplet mode) and  $0.18\text{-}0.32~\mu\text{m}$  (condensation mode) (Fig. 2). The coarse mode was primarily influenced by sea salt and dust, both of which originate from primary sources, while the droplet and condensation modes were largely driven by secondary processes such as cloud and fog interactions and gas-to-particle conversion (Seinfeld and Pandis, 2016).

In spring, the average concentrations of all three modes were relatively similar. However, in summer and autumn, the coarse-mode concentration was notably higher than other modes. During the winter cruise, the droplet mode became dominant. Using 1.8  $\mu$ m as the cutoff between coarse and fine PM, the fine PM concentrations were found to be 22.3  $\pm$  12.8  $\mu$ g m<sup>-3</sup> in spring and 18.7  $\pm$  7.5  $\mu$ g m<sup>-3</sup> in winter, both higher than coarse PM concentration (Fig. S1 and Table S1). This is different from the characteristics of pristine open-ocean boundary layers where coarse-mode sea salt typically dominates the total mass concentration (Russell et al., 2023). The fine PM concentration was 9.1  $\pm$  4.0  $\mu$ g m<sup>-3</sup> in summer, significantly lower than in other seasons. This seasonal variation aligns with the influence of the East Asian monsoon on air mass transport patterns, i.e., southeasterly winds from the cleaner oceanic environment dominate in summer, while northwesterly winds bring more pollution from the land in winter (Zhou et al., 2023). This pattern is also clearly supported by air mass backward trajectory analysis (Fig. S2). It is worth noting that although individual PM mass measurements, particularly during the autumn and winter campaigns, are subject to substantial uncertainties, the uncertainties associated with campaign-averaged values are much lower, making seasonal comparisons of the averages still reasonable.

Figure 2. Size distributions of aerosol mass concentration in four cruises. Each number represents a sample set and the black lines and error bars denote the means and corresponding standard deviations in each cruise.

# 3.2 Chemical composition

### 3.2.1 Water-soluble ions

The concentrations of water-soluble ions and trace elements in fine and coarse aerosols are listed in Table S1. As two major components of sea spray aerosol (SSA), Na<sup>+</sup> and Cl<sup>-</sup> were primarily found in coarse mode (87.4-95.4%, Figs. 3 and S1), consistent with previous understanding of the size distribution of SSA (Russell et al., 2023). Their concentrations peaked during the autumn cruise, approximately three times higher than in the other three seasons. This is primarily attributed to the enhanced SSA production driven by strong surface winds in autumn, consistent with the seasonal variations of Na<sup>+</sup> and Cl<sup>-</sup> previously observed at Huaniao Island (Zhou et al., 2021).

The three main secondary inorganic ions SO<sub>4</sub><sup>2-</sup>, NO<sub>3</sub><sup>-</sup> and NH<sub>4</sub><sup>+</sup> exhibited distinct size distributions and seasonal variations (Fig. 3). SO<sub>4</sub><sup>2-</sup> displayed a trimodal distribution. It is predominantly in condensation and droplet modes, but coarse mode also accounted for 11.9-27.1%, mainly from primary SO<sub>4</sub><sup>2-</sup> in sea salt (Fig. 3c). NO<sub>3</sub><sup>-</sup> exhibited two distinct size distribution patterns: one exclusively in the coarse mode and another with peaks in the condensation, droplet, and coarse modes (Fig. 3d). NH<sub>4</sub><sup>+</sup> generally exhibited a bimodal distribution (condensation and droplet modes), resulting from secondary reactions of gas-phase NH<sub>3</sub> (Fig. 3e).

 $K^+$  has multiple sources like biomass burning and SSA emission and exhibited a trimodal distribution. It was primarily found in fine particles in spring and winter, while coarse mode dominated in summer and autumn (Fig. S3).  $Mg^{2+}$  and  $Ca^{2+}$  were mainly associated with coarse particles (Fig. S3), contributed by sea salt and dust. However, in summer,  $Ca^{2+}$  showed another peak in the fine mode. The potential source for that is discussed in Section 4.3.2.  $C_2O_4^{2-}$  was predominantly present

in fine particles, formed through secondary reactions of anthropogenic and natural volatile organic compounds, including gaseous photochemical reactions and aqueous-phase processes in clouds (Guo et al., 2016).

Figure 3. Size distributions of (a) Na<sup>+</sup>, (b) Cl<sup>-</sup>, (c) SO<sub>4</sub><sup>2-</sup>, (d) NO<sub>3</sub><sup>-</sup>, and (e) NH<sub>4</sub><sup>+</sup> during four cruises. Each number represents a sample set and the thick black line with gray area below represents the average size distribution for each cruise (the same for Figs. 4 and 5).

# 3.2.2 Organic carbon

In this study, OC was not measured during the autumn and winter cruises, so the discussion here is based solely on the available data. As shown in Fig. S1 and Table S1, the total OC concentration was higher during the spring cruise (3.98  $\pm$  1.91  $\mu$ g m<sup>-3</sup>) compared to the summer cruise (1.66  $\pm$  0.87  $\mu$ g m<sup>-3</sup>). OC was primarily present in fine particles, but its size distribution varied significantly between the two seasons. During the spring cruise, OC was mainly concentrated in droplet mode, suggesting that secondary formation in the liquid phase was the dominant source (Fig. 4). In contrast, during the summer cruise, OC exhibited more diverse size distributions, with some samples showing a secondary peak in the coarse mode and others displaying high concentrations in particles smaller than 0.18  $\mu$ m. Potential reasons for the enrichment of OC in sub-0.18  $\mu$ m particles are discussed in Section 4.3.2.

Figure 4. Size distributions of OC during spring and summer cruises.

### 3.2.3 Trace elements

Among the trace elements measured in this study, nine (Al, Ba, Ca, Ce, Eu, Fe, Mg, Na, and Sr) were predominantly present in coarse particles across all four seasons (Table S1). The majority of Na was water-soluble Na<sup>+</sup> from sea salt, accounting for more than 80% of Na concentration. Al, Ba, Ce, Eu, and Fe are typical crustal elements primarily contributed by the long-range transport of dust, while Ca, Mg, and Sr have both sea salt and dust origins (Liu et al., 2022b; Pilson, 2012). These points are further supported by the SR-PMF results, as summarized in Section 3.3. The peak size of dust elements (e.g. Al) appeared at 3.2-5.6 μm, similar to that of sea salt (Fig. 5a). Concentrations of these dust elements increased significantly during strong dust transport events in spring and autumn cruises. For example, during October 23–24, 2020 (sample No. 11), the total aerosol Al concentration reached 3.72 μg m<sup>-3</sup>, approximately 7 times the autumn cruise average.

The concentrations of Co, Cu, K, Mn, and Rb were comparable in coarse and fine modes. Except for K, which has a sea salt origin, these elements in coarse particles mainly come from dust, while those in fine particles were predominantly from anthropogenic sources (Fu et al., 2008). Cd, Ge, Mo, Pb, Sb, Se, and Zn were primarily found in fine particles across all seasons (Fig. 5b-c, Table S1), indicating relatively low contributions from dust or sea salt. V and Ni are widely recognized as tracers for shipping emissions (Liu et al., 2017; Spada et al., 2018; Yu et al., 2021; Zhao et al., 2021). Their concentrations exhibited distinct size distributions and seasonal variations compared to other elements. During the spring and summer cruises, V and Ni were mostly found in fine particles smaller than 0.32 μm (Fig. 5d-e), consistent with the characteristics of primary aerosols from shipping emissions reported in previous studies. For instance, Zhang et al. (2020) found that OC/EC in ship exhaust aerosols was enriched in particles smaller than 0.43 μm. Zhou et al. (2019) and Winijkul et al. (2015) reported that freshly emitted ship particles peaked at ~0.15 μm, while the slightly larger peak size of V and Ni over the YECS (0.18 – 0.32 μm) may be due to particle growth via condensation and coagulation in the ambient atmosphere. During the autumn and winter cruises, the concentrations of fine-mode V and Ni dropped significantly, which were over an order of magnitude lower than those observed during the first two cruises (Table S1 and Fig. 5d-e). The reason for this decline and its potential environmental implications are further discussed in Section 4.4.

Figure 5. Size distributions of (a) Al, (b) Cd, (c) Se, (d) V, and (e) Ni during four cruises.

# 230 3.3 Overview of the SR-PMF source apportionment results

235

250

The SR-PMF model was run with factor numbers ranging from 4 to 9. Based on the trend of  $Q/Q_{exp}$  with increasing factor numbers (Fig. S4a-b) and the physical interpretability of the solutions, we determine that the 7-factor solution provides the most reasonable results. The PMF-reconstructed concentrations show strong agreement with the measured values for all species, with  $R^2 > 0.90$  for 29 out of 32 species and  $R^2 > 0.78$  for the remaining 3 (Table S2 and Fig. S4c). The normalized contribution intensity of each PMF factor across different size ranges in each sample set is shown in Fig. S5. The average contributions to the size-integrated concentrations of different components (i.e., the chemical composition profiles) and the mean normalized size distributions of each factor are presented in Fig. 6 and Fig. 7, respectively. The attribution of each PMF factor to specific source or process is based on both its chemical composition profile and size distribution, as explained in detail below.

Factor 1 is characterized by high loadings of crustal elements such as Al, Ba, Ca, Ce, Eu, and Fe, accounting for more than 60% of their total concentrations (Figs. 6 and S6). This factor clearly represents the dust source, with a unimodal size distribution peaking at 3.2 – 5.6 μm. Factor 2 contributes more than 50% to Cd, Pb, and Zn, over 30% to Cu, Mn, Mo, and Sb, and has minimal contributions to secondary ions. This factor corresponds to primary anthropogenic emissions on land such as industrial processing and transportation. The aerosols from this source are predominantly in the fine size range (i.e., <1.8 μm), though the relative contribution of coarse particles is non-negligible (22.3%).

Factors 3 and 4 both make significant contributions to the secondary ions NH<sub>4</sub><sup>+</sup> and SO<sub>4</sub><sup>2-</sup> and exhibit similar size distributions which are concentrated in small particles (90.2% and 88.6%). However, their species profiles are different. For example, Factor 3 is a major source of NO<sub>3</sub><sup>-</sup>, whereas Factor 4 makes no contribution to it. In addition to secondary ions, Factor 3 contributes over 30% to K<sup>+</sup> (a biomass burning indicator) and trace elements such as Ge, Pb and Sb. In contrast, Factor 4 has a much lower contribution to most trace elements but contributes significantly to C<sub>2</sub>O<sub>4</sub><sup>2-</sup> (88.1%) and Se (40.0%). Based on these characteristics, we attribute Factor 3 to continental secondary and burning-related sources (abbreviated as CS&B), while Factor 4 represents aged continental and marine secondary sources (abbreviated as AC&MS). More details are discussed in Section 4.1.

Factor 5 is characterized by high contributions to Ni and V, two well-known tracers for shipping emissions (Yu et al., 2021; Zhao et al., 2021). This factor is therefore attributed to aerosols from shipping emission. Factors 6 and 7 are two sea-salt-related sources, contributing significantly to sea-salt cations (e.g., Na<sup>+</sup>, K<sup>+</sup>, Mg<sup>2+</sup>, Ca<sup>2+</sup>, and Sr<sup>2+</sup>) and predominantly appearing in the coarse mode (Fig. 7). However, Factor 6 accounts for 94.6% of Cl<sup>-</sup> and 0% of NO<sub>3</sub><sup>-</sup>, while Factor 7 contributes 0% to Cl<sup>-</sup> but 43.2%% to NO<sub>3</sub><sup>-</sup>. These characteristics indicate that Factor 6 represents fresh sea salt, whereas Factor 7 corresponds to aged sea salt. More details are discussed in Section 4.3.

Figure 6. The mean absolute contribution (blue bars) and relative contribution (red squares) of each PMF factor to the size-integrated concentrations of different aerosol components over the eastern China seas across all four seasons.

Figure 7. Mean size distributions of the normalized contribution intensity of different PMF factors across all four seasons.

### 265 4 Discussion

270

# 4.1 Continental anthropogenic pollution

### 4.1.1 Aging of continental secondary aerosols and nitrate redistribution

In polluted continental environments with abundant NO<sub>x</sub> and NH<sub>3</sub> in the atmosphere, NO<sub>3</sub><sup>-</sup> primarily exists in the form of NH<sub>4</sub>NO<sub>3</sub> and resides in fine particles (Guo et al., 2010), which is captured by the CS&B factor in our SR-PMF analysis. However, NH<sub>4</sub>NO<sub>3</sub> is thermodynamically unstable and can decompose into gaseous HNO<sub>3</sub> and NH<sub>3</sub> under warm temperatures or when the ambient concentrations of HNO<sub>3</sub> and NH<sub>3</sub> are low (Guo et al., 2010; Uno et al., 2017):

$$NH_4NO_3 \rightleftharpoons HNO_3(g) + NH_3(g)$$
 (1)

As continental air masses are transported over the ocean, the concentrations of gaseous HNO<sub>3</sub> and NH<sub>3</sub> decline rapidly due to the substantially lower emission fluxes of NO<sub>x</sub> and NH<sub>3</sub> in marine environments. This shifts the gas-particle equilibrium toward the gas phase, resulting in the gradual decomposition of fine-mode NH<sub>4</sub>NO<sub>3</sub>.

In the presence of mineral dust or SSA particles, gaseous HNO<sub>3</sub> can be taken up and undergo heterogeneous reactions with specific components such as CaCO<sub>3</sub> and NaCl, leading to the formation of thermodynamically stable nitrate compounds like Ca(NO<sub>3</sub>)<sub>2</sub> and NaNO<sub>3</sub> as shown below (Usher et al., 2003; Rossi, 2003; Bondy et al., 2017; Liu et al., 2022a). In addition to HNO<sub>3</sub>, dust and SSA aerosols can also uptake other nitrogen-containing acidic gases (e.g., N<sub>2</sub>O<sub>5</sub>, NO<sub>2</sub>), which similarly undergo heterogeneous reactions to form stable nitrates (Rossi, 2003; Tang et al., 2016). For SSA, these reactions also result in the conversion of Cl<sup>-</sup> to volatile chlorine-containing gases (e.g., HCl, ClNO<sub>2</sub>), causing sea salt chloride depletion.

$$2HNO_3(g) + CaCO_3 \rightarrow Ca(NO_3)_2 + CO_2(g) + H_2O$$
 (2)

$$HNO_3(g) + NaCl \rightarrow NaNO_3 + HCl(g)$$
 (3)

280

290

295

$$N_2O_5(g) + NaCl \rightarrow NaNO_3 + ClNO_2(g)$$
 (4)

Collectively, these heterogeneous reactions further reduce the atmospheric concentration of gaseous HNO<sub>3</sub>, thus promoting the decomposition and inhibiting the reformation of fine-mode NH<sub>4</sub>NO<sub>3</sub>. Since dust and SSA particles are mainly distributed in the coarse mode, the resulting nitrate products are likewise primarily found in coarse particles. This leads to a redistribution of NO<sub>3</sub><sup>-</sup> from the fine to the coarse mode.

This NO<sub>3</sub><sup>-</sup> redistribution mechanism, along with the aging of continental secondary aerosols, is supported by our observations. As shown in Fig. 8, the concentration ratio of fine-mode to coarse-mode NO<sub>3</sub><sup>-</sup> decreases significantly with increasing time that the 72-hour air mass backward trajectories spent over the ocean. Specifically, when air masses had travelled over the ocean for less than 24 hours—i.e., when continental aerosols were relatively fresh—the median fine-to-coarse ratio of NO<sub>3</sub><sup>-</sup> was 2.23 (Fig. 8b), indicating that in more than half of the samples, over two-thirds of the NO<sub>3</sub><sup>-</sup> mass remained in fine particles. In contrast, when the time over ocean exceeded 48 hours (i.e., the continental aerosols were highly aged), the fine-to-coarse ratio fell below 1 in 92.6% of the samples (even below 0.15 in 59% of samples). Although an extreme outlier appears in the top-right corner of Fig. 8a (time over ocean = 71.8 hours, fine-to-coarse ratio = 4.0; Sample 5 from the winter campaign), this anomaly is explainable because the research vessel passed through the Zhoushan archipelago which hosts the world largest port locates and has a relatively high population density, and was likely strongly influenced by anthropogenic pollution.

Beyond the time over ocean, the fine-to-coarse ratio of NO<sub>3</sub><sup>-</sup> is also strongly influenced by SSA abundance (represented by Na<sup>+</sup> concentration here). As shown in Fig. 8a, at a given time over ocean level, samples with higher Na<sup>+</sup> concentrations generally exhibited lower fine-to-coarse NO<sub>3</sub><sup>-</sup> ratios. In addition, during the most intense dust transport event in this study (22 October – 24 October 2020, marked by the two filled circles in Fig. 8a), the observed fine-to-coarse ratios were significantly lower than typical values at similar levels of time over ocean and Na<sup>+</sup> concentrations. These results further support the role of SSA and dust in facilitating the transformation of NO<sub>3</sub><sup>-</sup> from fine to coarse mode via heterogeneous reactions. Furthermore, air temperature also appears to influence NO<sub>3</sub><sup>-</sup> partitioning. As shown in Fig. 8a, higher temperatures

are typically associated with lower fine-to-coarse ratios. When the average air temperature along the backward trajectory exceeded 15 °C, all samples exhibited fine-to-coarse ratios below 1, except for two cases with very low Na<sup>+</sup> concentrations.

Therefore, the fine-mode fraction of  $NO_3^-$  is generally associated with the transport time (i.e., the degree of aging) of continental secondary aerosols over the ocean. A high fine-mode fraction of  $NO_3^-$  in marine aerosols, along with a high contribution intensity of CS&B factor, suggests a pronounced and relatively direct influence of terrestrial transport before the aerosols have fully aged. This could be due to either severe atmospheric pollution on land being transported to the ocean or sampling conducted in regions close to the coastline. Across all four cruise campaigns, 49.1% (26/53) of the aerosol samples were identified as strongly influenced by this process, based on the following three criteria: (1) there is a prominent accumulation-mode or droplet-mode peak in the  $NO_3^-$  size distribution, defined as  $\frac{[NO_3^-]_{0.18-0.32}}{\frac{1}{2}\times([NO_3^-]_{0.10-0.18}+[NO_3^-]_{0.32-0.56})} > 1$  or

 $\frac{[NO_3^-]_{0.56-1.0}}{\frac{1}{2}\times([NO_3^-]_{0.32-0.56}+[NO_3^-]_{1.0-1.8})} > 1; (2) \text{ more than } 15\% \text{ of the } NO_3^- \text{ concentration is distributed in particles with a diameter smaller than } 1.8 \ \mu\text{m} \text{ (PM}_{1.8}); \text{ and } (3) \text{ the CS&B factor contributes more than } 15\% \text{ of the total } NO_3^- \text{ concentration. } NO_3^- \text{ in these samples exhibits bimodal or trimodal size distributions (Fig. S7). In other words, the continental secondary aerosols in 50.9% of all samples were highly aged, with <math>NO_3^-$  predominantly existing in coarse particles. In summer, as air masses primarily originate from the southeastern ocean (Fig. S2), the direct influence of terrestrial transport and thus the overall contribution from CS&B is significantly weaker (Fig. 9). Correspondingly, the continental input of fine-mode  $NH_4NO_3$  is limited, and any continental aerosols present are likely highly aged due to extended transport times over the ocean. Higher temperatures in summer may further promote the decomposition and aging of fine-mode  $NH_4NO_3$ . As a result, except for the first two samples influenced by sea fog,  $NO_3^-$  in summer all exhibits a unimodal distribution in the coarse mode (Fig. S7). In contrast,  $SO_4^{2-}$  does not undergo such thermodynamically driven fine-to-coarse transformation and remains in the fine mode even after aging, owing to its chemical stability and low volatility under ambient conditions.

The shift in NO<sub>3</sub><sup>-</sup> size distribution from fine mode to coarse mode has significant implications for its environmental effects. For example, it will reduce the mass scattering efficiency of NO<sub>3</sub><sup>-</sup>, thereby weakening its direct radiative effect (Chen et al., 2020). Moreover, it increases the deposition velocity and decreases the atmospheric residence time of NO<sub>3</sub><sup>-</sup> (Chen et al., 2020; Zhu et al., 2013), which in turn significantly limits its transport distance. Consequently, the spatial extent of oceanic regions influenced by NO<sub>3</sub><sup>-</sup> deposition is also reduced, potentially altering its ecological effects.

Figure 8. (a) Scattering plot between the concentration ratio of fine-mode to coarse-mode  $NO_3^-$  (Fine-to-coarse ratio<sub>[NO\_3^-]</sub>) and the time that 72-hour air mass backward trajectories spent over the ocean (Time over ocean). The color and size of the scatter points indicate the average air temperature along the trajectory (Temperature<sub>traj</sub>) and the size-integrated  $Na^+$  concentration, respectively. The two filled circles highlight cases observed during the strongest dust event (22 October – 24 October 2020). (b) Boxcharts of the Fine-to-coarse ratio<sub>[NO\_3^-]</sub> grouped by different ranges of Time over ocean. The boxes represent the interquartile range (25th to 75th percentiles), the horizontal lines indicate the median, and the whiskers represent the highest and lowest values within median  $\pm$  1.5 interquartile range. Red circles denote the mean values, and stars indicate statistical outliers.

# 4.1.2 Mixing with marine secondary aerosols and contribution to PM<sub>1.8</sub>

When continental secondary aerosols are transported over ocean, in addition to undergoing evaporation of NH<sub>4</sub>NO<sub>3</sub>, they can mix with marine secondary aerosols both externally and internally. Marine secondary components are produced via the oxidation of marine reactive gases such as dimethly sulfide (DMS), isoprene, and monoterpenes. In this study, the joint contribution of aged continental secondary aerosols and marine secondary aerosols was effectively resolved by SR-PMF (represented by Factor 4, AC&MS).

It is worth noting that, in addition to the commonly studied DMS, marine microorganisms also emit volatile Se-containing gases, such as dimethyl selenide (DMSe) and dimethyl diselenide (DMDSe), which are often positively correlated with DMS (Amouroux et al., 2001). These Se-containing gases are oxidized in the atmosphere to form low-volatile species (e.g.,  $H_2SeO_3$  and  $H_2SeO_4$ ) and enter into the particulate phase (Wen and Carignan, 2007). As a result, AC&MS contributes significantly to the observed Se concentrations in marine aerosols (Fig. 6), explaining why Se concentrations do not show the typical drastic summer decline observed for other terrestrial trace elements (Table S1). Additionally, both continental and marine environments emit precursors of  $C_2O_4^{2-}$ , such as isoprene and terpenes from the forests, acetylene and aromatics from

anthropogenic sources, and isoprene and unsaturated fatty acids from the ocean surface (Myriokefalitakis et al., 2008; Kawamura and Sakaguchi, 1999; Cui et al., 2023; Boreddy et al., 2017; Myriokefalitakis et al., 2011). These precursors can undergo photochemical oxidation to form small organic acids and carbonyl compounds, which are further oxidized to C<sub>2</sub>O<sub>4</sub><sup>2</sup>- via aqueous-phase reactions (Myriokefalitakis et al., 2011; Carlton et al., 2007; Myriokefalitakis et al., 2008). Previous studies have highlighted a mixing contribution of marine and continental origins to C<sub>2</sub>O<sub>4</sub><sup>2</sup>- in coastal regions (Wang et al., 2016; Zhou et al., 2015), which is also consistent with our SR-PMF results.

Summing the contributions of each PMF factor to the total concentrations of the chemical species included in the SR-PMF analysis, we find that the combined contribution from CS&B and AC&MS factors is strongly correlated with the PM<sub>1.8</sub> mass concentration (R<sup>2</sup> = 0.678; Fig. S8a). It is important to note that organics were not included in the SR-PMF analysis due to the lack of measurement during the autumn and winter campaigns. Consequently, the PMF-reconstructed concentrations are significantly lower than the observed PM<sub>1.8</sub> mass concentrations, with a linear regression slope of 0.366 (Fig. S8a). Including other PMF factors only slightly improves the regression slope, from 0.366 to 0.384 (Fig. S8). Organic aerosols are known to be abundant in fine particles over eastern China, with concentrations comparable to those of inorganic components, and they predominantly originate from combustion-related sources and secondary formation processes (Daellenbach et al., 2024; Huang et al., 2014). Therefore, it is likely that CS&B and AC&MS represent the primary sources of fine particles in our study, especially during spring, autumn, and winter.

The autumn campaign was conducted after the outbreak of COVID-19 pandemic, during which anthropogenic emissions may have been reduced by control measures such as lockdowns. However, the most stringent lockdowns and associated emission reductions in East Asia occurred in the first half of 2020. By the second half of the year, emissions had largely returned to 2019 levels, with relative differences below 10% (Doumbia et al., 2021; Zheng et al., 2021). Therefore, although some influence from COVID-related measures cannot be ruled out, they are not expected to have significantly affected our observations.

# 4.2 Dust aerosol and its aging

The transport of East Asian dust to the marginal seas of the northwest Pacific often passes through the densely populated regions of northern and eastern China. As a result, significant increases in dust aerosol concentrations over the eastern China seas are often accompanied by notable rises in continental pollution aerosols. In this study, three strong co-transport events of dust and continental pollution aerosols (27 March – 28 March 2017, Sample 1; 14 April 2017, Sample 8 and 9; 22 October – 24 October 2020, Sample 10 and 11) were observed, as denoted by the events T1 – T3 in Fig. 9. These events are characterized by concurrent strong increases in the contribution intensity of three PMF factors: dust, primary anthropogenic emissions, and CS&B. Backward trajectory analyses (Fig. S9) indicate that the air masses associated with these events originated from the Gobi Desert in Mongolia and passed through the densely populated regions of northern and eastern China. The two samples corresponding to event T3 were collected at the same location near the coast. In this event, peaks in

anthropogenic pollutant concentrations and PMF source contributions from primary anthropogenic emissions and CS&B appeared in Sample 10, preceding the peak in dust concentration observed in Sample 11 (Fig. 9). This temporal pattern is consistent with the dynamics of long-range transport, where pollutants from continental China typically arrive before dust originating from more distant inland regions. Local anthropogenic emissions in the coastal area played a secondary role in shaping the observed aerosol concentrations and properties during this event.

During transport, dust aerosols mix and react with anthropogenic pollutants, undergoing chemical aging process. As discussed above, dust can participate in heterogeneous reactions with nitrogen-containing acidic gases, leading to the formation of coarse-mode  $NO_3^-$ . Based on SR-PMF analysis, the average concentration of  $NO_3^-$  associated with dust was 0.43  $\mu$ g m<sup>-3</sup>, accounting for 12.7% of the total  $NO_3^-$  concentration across all four cruises (Figs. 6 and S6, Table S3). During intense dust transport events,  $NO_3^-$  from dust aging can constitute more than half of the total  $NO_3^-$  concentration. For instance, in sample 11 of the autumn cruise, the  $NO_3^-$  concentration in aged dust was 4.68  $\mu$ g m<sup>-3</sup>, making up 59.5% of the total  $NO_3^-$  concentration.

This chemical aging process of dust aerosols has several important environmental implications. First, the formation of a water-soluble coating on aged dust particles increases their hygroscopicity, enhancing their role as cloud condensation nuclei (CCN) (Gibson et al., 2007; Tang et al., 2016). Second, the chemical aging increases aerosol acidity, potentially enhancing the solubility of metal elements in dust (Gaston, 2020). As dissolved metals and NO<sub>3</sub><sup>-</sup> are bioavailable, this process may amplify the overall ecological impacts of dust deposition on phytoplankton in the northwestern Pacific.

Figure 9. Temporal variations of the normalized contribution intensity of different sources and the mass concentrations of fine- and coarse-mode aerosols. The x-axis label represents the sample ID in chronological order within each cruise. T1 – T3 with brown shaded bands denote the three co-transport events of dust and continental pollution events.

### 4.3 Sea spray aerosol and its aging

### 4.3.1 Inorganic sea salt and chloride depletion

SSA is a major contributor to aerosol mass concentration in the marine boundary layer, primarily existing in coarse particles and dominated by inorganic sea salt components such as Na<sup>+</sup> and Cl<sup>-</sup>. In this study, the SR-PMF analysis was conducted without imposing any prior constraints on sea salt composition ratios. Nevertheless, it successfully resolved two distinct factors: fresh (unaged) sea salt and aged sea salt. The molar ratio of Cl<sup>-</sup> to Na<sup>+</sup> in the fresh sea salt factor is 1.18, closely matching the theoretical seawater value of 1.16 (Pilson, 2012), supporting the robustness of the results. In the aged sea salt factor, Cl<sup>-</sup> concentration is completely depleted, and NO<sub>3</sub><sup>-</sup> becomes the dominant anion, accounting for 87.0% of the equivalent concentration. After subtracting sea salt-derived SO<sub>4</sub><sup>2-</sup> based on the seawater SO<sub>4</sub><sup>2-</sup>/Na<sup>+</sup> molar ratio of 0.058 (Keene et al., 2007), NO<sub>3</sub><sup>-</sup> comprises 97.1% of the remaining anions, indicating that almost all chloride depletion is attributed to the reactions with nitrogen-containing gases rather than sulfur-containing gases. It is important to note that in the real atmosphere, SSA particles are typically only partially aged. An SSA particle can be viewed as an internal mixture of unreacted sea salt components and those in which Cl<sup>-</sup> has been replaced by NO<sub>3</sub><sup>-</sup>, which are represented by the fresh sea salt and aged sea salt PMF factors, respectively. In other words, these two PMF factors do not represent an external mixture of entirely fresh and entirely aged SSA particles. The combined contributions of these two factors reflect the actual aging state of SSA in the atmosphere.

As noted in Section 4.1, the heterogeneous reactions between SSA and nitrogen-containing acidic gases shift NO<sub>3</sub><sup>-</sup> from smaller to larger particle sizes (Chen et al., 2020), greatly affecting its atmospheric lifetime and transport distance. Based on the SR-PMF results, the average concentration of NO<sub>3</sub><sup>-</sup> associated with aged SSA was 1.47 μg m<sup>-3</sup>, accounting for 43.2% of the total NO<sub>3</sub><sup>-</sup> concentration (Figs. 6 and S6, Table S3). This contribution is 3.4 times greater than that with dust (i.e., represented by dust factor, 0.43 μg m<sup>-3</sup>), indicating that SSA plays a more significant role in the size redistribution of NO<sub>3</sub><sup>-</sup>. During summer, nearly all NO<sub>3</sub><sup>-</sup> mass concentration (95.3%) was associated with aged SSA. Summing the concentration of all species contributed by fresh and aged SSA factors, their combined average concentration accounts for 57.2% of the average PM<sub>>1.8</sub> mass. This proportion would likely be even higher if components not included in the SR-PMF analysis were considered, indicating SSA generally dominates the coarse-mode aerosol concentration. However, during strong dust transport events such as 22 October – 24 October 2020 (T3 in Fig. 9), dust aerosols can become the primary contributor to coarse-mode aerosol mass. By combining the PMF results with a dust equation that convert trace elements to their common oxide forms (Liu et al., 2022b), the coarse-mode dust aerosol concentration was estimated at 32.1 μg m<sup>-3</sup>, accounting for 65.9% of the PM<sub>>1.8</sub> concentration during this period.

Notably, the particle size distributions of fresh and aged SSA exhibit slight differences. For fresh sea salt factor, the concentration in the 3.2–5.6  $\mu$ m size range is only slightly higher than in the 5.6–10  $\mu$ m range, and the concentration in particles larger than 18  $\mu$ m exceeded that in the 10–18  $\mu$ m range (Fig. 7). In contrast, aged sea salt shows a monotonic increase in concentration from >18  $\mu$ m to 3.2–5.6  $\mu$ m, with significantly higher concentrations in the 3.2–5.6  $\mu$ m size range

than in adjacent sizes. Consequently, the mass median diameter of aged sea salt is smaller than that of fresh sea salt. This difference is linked to the surface-area dependence of heterogeneous reaction rates between SSA particles and acidic gases, as gas uptake onto aerosols is proportional to aerosol surface area and often limits the overall reaction rate (Zhuang et al., 1999; Rossi, 2003). As shown in Fig. S10, the normalized size distribution of [Fresh sea salt]/D<sub>p</sub> (a proxy for the surface area of fresh sea salt) closely resembles the size distribution of the aged sea salt factor, supporting the surface-area-limited nature of the heterogeneous reaction mechanism.

Similarly, as smaller particles have a higher specific surface area, leading to faster reaction rates per unit mass, an inverse relationship between chloride depletion ratio and SSA particle size has been reported by previous studies (Zhuang et al., 1999; Yao et al., 2003). This trend was observed in this study as well, as shown in Fig. 10 (data for  $D_p < 1$   $\mu$ m were excluded due to low concentrations and high measurement uncertainties of sea salt components). Here the chloride depletion ratio ( $Cl_{dep}$ ) was calculated based on the measured molar concentrations of  $Cl^-$  and  $Na^+$ :  $Cl_{dep} = 1 - [Cl^-]/(1.16 \times [Na^+])$ , where 1.16 is the abovementioned theoretical molar ration between  $Cl^-$  and  $Na^+$  in fresh sea salt aerosols, consistent with seawater composition. The cruise-averaged chloride depletion ratio increased from 3.5%–14.9% in particles >18  $\mu$ m to 36.7%–61.4% in particles between 1 and 1.8  $\mu$ m. Across all four cruises, the average chloride depletion ratio was 27.7%, peaking in summer (40.8%). The depletion ratio reached its lowest in autumn (20.6%), likely due to higher SSA emission fluxes and limited availability of nitrogen-containing acidic gases.

HCl and ClNO<sub>2</sub> released from SSA aging serves as an important source of reactive chlorine in the atmosphere, which has been shown to enhance ozone formation in polluted regions (Li et al., 2021; Wang et al., 2020). In addition to chloride, bromide in SSA can undergo similar processes, releasing reactive bromine gases and further generating bromine-containing radicals (Parrella et al., 2012; Sander et al., 2003). Collectively, these reactive halogen gases from sea salt aging over coastal seas may influence ozone pollution in eastern China, though the extent of this influence requires further investigation.

450

455

Figure 10. The average size distributions of Cl<sup>-</sup> depletion percentage in particles larger than 1 μm in different cruise campaigns.

# 4.3.2 Organic carbon and calcium enrichment

In addition to the inorganic sea salt components in coarse particles, SSA generation also introduces organic matter from the sea surface into the atmosphere, leading to the internal mixing of organics within SSA, particularly in fine particles (Bertram et al., 2018; Quinn et al., 2015; O'dowd et al., 2004). The eastern China seas are nutrient-rich and highly productive, with elevated concentrations of phytoplankton biomass and dissolved organic matter (He et al., 2013; Hung et al., 2003). As a result, SSA in this region may contain substantial organic content. During the summer cruise, high concentrations of OC were observed in fine particles smaller than 0.18 μm, particularly from 12 July to 16 July (corresponding to samples 7–8) following Typhoon Maria's passage (Fig. 4). Given the reduced influence of terrestrial sources in summer and the distinct size distribution from typical condensation-mode particles, these fine-mode OC are likely SSA-derived.

Additionally, the summer campaign revealed a unique  $Ca^{2+}$  size distribution, with not only a coarse mode from dust and inorganic sea salt but also a prominent peak in the  $0.18-0.32~\mu m$  range, where fine-mode  $Ca^{2+}$  accounted for ~40% of the total concentration (Fig. S3c). Since  $Ca^{2+}$  cannot form secondarily from gaseous precursors, this unusual size distribution may also be linked to SSA. Previous studies suggest that along with the enrichment of saccharides in SSA, divalent cations like  $Ca^{2+}$  may chelate with these organic compounds and become enriched in SSA as well (Jayarathne et al., 2016; Salter et al., 2016). Notably, the peak size of  $Ca^{2+}$  in fine particles did not align with OC peak in particles smaller than 0.18  $\mu m$ , possibly due to substrate selectivity in  $Ca^{2+}$  chelation, with chelating organic compounds primarily concentrated in the 0.18–0.32  $\mu m$  range. Research on this topic in the eastern China seas remains limited, and this study provides only preliminary insights based on a small sample set. More extensive and comprehensive chemical composition analyses are needed in the future to confirm these findings.

# 4.4 Shipping emissions

Aerosols from shipping emissions are primarily distributed in the particle size range below 0.32 μm (Fig. 7). The SR-PMF analysis reveals that this source contributes 84.8% and 60.1% of the total V and Ni, respectively. Coarse-mode V and Ni are mainly attributed to dust, accounting for 8.6% and 15.9% of their total concentrations, while primary anthropogenic emissions contribute an additional 12.8% to Ni, primarily in the size range of 0.56–3.2 μm. During the spring and summer cruises, the average concentrations of V and Ni in PM<sub>1.8</sub> were 12.63 and 8.84 ng m<sup>-3</sup>, respectively. In contrast, during the autumn and winter cruises, these values dropped to 0.64 and 1.11 ng m<sup>-3</sup>, representing decreases of 94.5% and 87.4%, respectively (Fig. 5 and Table S1). For particles smaller than 0.32 μm, the V and Ni concentrations during autumn and winter were 0.39 and 0.50 ng m<sup>-3</sup>, compared to 9.05 and 4.08 ng m<sup>-3</sup> in spring and summer, with reductions of 95.7% and 87.8%, respectively. The reduction in shipping emission due to the COVID-19 pandemic is not the primary driver for this

phenomenon, as the estimated decrease in East Asia was only within 20% (Yi et al., 2024), and the winter campaign was conducted before the pandemic significantly impacted ship activity.

This significant decline in V and Ni concentrations is likely linked to improvement in marine fuel quality. According to the International Maritime Organization's (IMO) 2020 regulation, the sulfur content of marine fuels was reduced globally from 3.5% to 0.5%, effective 1 January 2020. A side effect of this regulation is the reduced concentration of certain trace metals in marine fuels (Yu et al., 2021; Moreira et al., 2024). The winter cruise of this study began on 26 December 2019, with the first sample set collected before 1 January 2020. Our measurements show that V and Ni concentrations corresponding to this sample were already significantly lower than in the previous cruises, suggesting that most ships in the studied area had switched to the new standard fuel before the IMO 2020 regulation came into effect.

Similar reductions in V and Ni concentrations in aerosols due to improved marine fuel quality have been reported in other studies (Spada et al., 2018; Yu et al., 2021; Moreira et al., 2024). At the port of Paranaguá in Brazil, V and Ni concentrations in PM<sub>2.5</sub> decreased by 86.2% and 62.1%, respectively, from 2019 to 2020 following the enforcement of the IMO 2020 regulation (Moreira et al., 2024). Yu et al. (2021) observed a decrease of 89% in PM<sub>2.5</sub> V concentrations over Shanghai in 2020 compared to 2017, while Ni levels dropped by only 40%. Their analysis of marine fuel samples confirmed a significant decline in V content, but the reduction in Ni was much smaller. However, in this study, the reductions in fine-mode Ni concentrations over the eastern China seas after 2020 were also substantial (>85%), though slightly less than those of V. This discrepancy may arise from two factors: (1) Shanghai is located within China's Domestic Emission Control Areas (DECAs), where the use of low-sulfur fuel has been mandatory since 2017, leading to prior reductions in V and Ni; and (2) Ni has a larger fraction of contributions from non-ship-related sources than V (e.g., primary anthropogenic emissions and dust), resulting in higher background Ni concentrations over urban areas than over the ocean, and these background concentrations remain unaffected by fuel quality changes. Moreover, the fuel samples analyzed by Yu et al. (2021) may not fully represent the fuels commonly used by marine ships, as Ni content in fuel has likely decreased more significantly than their findings suggest.

In this study, the Ni/V ratio in the source profile of shipping emissions derived from SR-PMF analysis is 0.42. For particles smaller than 0.32 µm, V and Ni concentrations show a strong correlation during the spring and summer cruises, with a linear regression slope of approximately 0.37, close to the PMF results. After the implementation of the IMO 2020 regulation, the slope during the autumn and winter cruises increased to 2.09 (Fig. 11). These values, both before and after IMO 2020 regulation, are in close agreement with those reported by Yu et al. (2021). It should be noted that the use of orthogonal distance regression (ODR) for linear fitting is more appropriate than ordinary least squares (OLS) fitting in this context, as the measurements of V and Ni have similar uncertainties. This is also reflected in the greater consistency of ODR results across different campaigns under the same marine fuel condition, whereas OLS yields significantly different slopes for autumn (0.54) and winter (1.23) cruises. These findings suggest that the Ni/V ratio in shipping emissions has undergone a notable shift following fuel quality improvements. The PMF analysis in this study combines data from all cruise campaigns

and produces a single fixed source profile for shipping emission. However, this source profile is largely determined by the high concentrations observed during the spring and summer cruises. Thus, the PMF-based quantification of shipping emissions for the autumn and winter campaigns may be less accurate. From the size distribution of V, it is clear that a significant ship-emission mode persists in the sub-0.32 µm size range during the autumn and winter cruises. This mode differs from the condensation mode of non-ship sources, but it aligns with the V size distribution observed in the spring and summer (Fig. 5). Therefore, despite the significant decline in V concentrations from shipping emissions, making it no longer the primary source of bulk aerosol V, V in the sub-0.32 µm size range can still serve as an indicator for shipping emissions to perform source apportionment. However, this requires targeted sampling strategies for fine aerosols and high-precision detection methods, and the changes in the source profile must also be taken into account.

The SR-PMF analysis also highlights that shipping emission is a significant source of SO<sub>4</sub><sup>2-</sup> in fine aerosols. During the 535 spring and summer cruises, shipping emissions contributed, on average, 21.1% of the total SO<sub>4</sub><sup>2-</sup> concentration. Notably, in the summer, when terrestrial influences were minimal, the contribution reached 35.4%, with 45.6% for fine  $SO_4^{2-}$  (< 1.8 µm) and up to 65.5% for SO<sub>4</sub><sup>2-</sup> in aerosols smaller than 0.32 μm. Thus, although shipping emissions may have contributed less to the total aerosol mass compared to other sources, their significant impact on the CCN population (dominated by particles < 540 300 nm) likely had considerable regional climate implications before 2020. Following the IMO 2020 regulation, the contribution of shipping emissions to  $SO_4^{2-}$  became negligible, consistent with the reduction in  $SO_x$  emissions due to lower sulfur content in fuels. However, as mentioned earlier, the source profile of shipping emissions has also changed, which may affect the accuracy of the quantified contribution to SO<sub>4</sub><sup>2-</sup>. In addition, here the PMF-resolved SO<sub>4</sub><sup>2-</sup> from shipping emission likely reflects primarily the direct emission of SO<sub>4</sub><sup>2-</sup> in the ship exhaust. The secondary SO<sub>4</sub><sup>2-</sup> formed from the ship-emitted 545 SO<sub>2</sub> may exhibit a different size distribution and may not be fully captured by the PMF shipping emission factor. Consequently, the total contribution of shipping emission to SO<sub>4</sub><sup>2-</sup> could be underestimated in our results. Further observations and modeling studies are needed to better address these uncertainties and understand the effects of this marine fuel regulation on regional air quality and climate.

Figure 11. Correlations between the concentrations of Ni and V in (a) spring, (b) summer, (c) autumn, and (d) winter cruises. Different markers represent the concentrations in different size ranges. The gray and red dashed lines correspond to the ODR and OLS fitting for the concentrations in the  $0.056 - 0.32 \mu m$  size range. In panel (a), the OLS fitting line is not shown as it closely overlaps with the ODR fitting line.

# 4.5 Summary of NO<sub>3</sub><sup>-</sup> and SO<sub>4</sub><sup>2-</sup> sources across particle sizes

This section summarizes the contributions of different sources to NO<sub>3</sub><sup>-</sup> and SO<sub>4</sub><sup>2</sup><sup>-</sup> across different particle sizes. Consistent with the previous discussion, dust and aged sea salt aerosol (SSA) are the two dominant contributors to coarse-mode NO<sub>3</sub><sup>-</sup> over the eastern China seas. This reflects the importance of heterogeneous reactions between nitrogen-containing acidic gases and dust or SSA particles. On average across all four seasons, these two pathways together accounted for more than 85% of NO<sub>3</sub><sup>-</sup> in all size bins larger than 1.8 μm (Fig. 12a). Specifically, dust contributed 17% to 39% to NO<sub>3</sub><sup>-</sup> in different size bins of coarse particles, with an overall contribution of 20.5%. Aged SSA contributed a higher proportion (72.0%), ranging from 52% to 75% in coarse particles and exceeding 70% in the 1.8–10 μm range. This strong contribution of aged SSA to

coarse-mode NO<sub>3</sub><sup>-</sup> was consistent across all seasons, and particularly prominent in summer, when aged SSA accounted for 95.4% of total coarse-mode NO<sub>3</sub><sup>-</sup> (Fig. S11). Continental secondary formation was the dominant source of fine-mode NO<sub>3</sub><sup>-</sup>, accounting for over 90% of the all-season average NO<sub>3</sub><sup>-</sup> concentration in particles smaller than 1.0 µm (Fig. 12a). This pattern held for the seasonal averages in spring, autumn, and winter (Fig. S11). While in summer, due to the extremely low concentrations and high measurement uncertainties, the PMF results for submicron NO<sub>3</sub><sup>-</sup> may not be statistically robust.

565

570

Sea salt (fresh SSA + aged SSA) contributed 56.8% of all-season average coarse-mode  $SO_4^{2-}$  concentration, and this relative contribution peaked at 5.6 – 10 µm (73.1%, Fig. 12b). In autumn, when SSA concentrations were highest, the contribution to coarse-mode  $SO_4^{2-}$  reached 70.9%. It is worth noting that  $SO_4^{2-}$  associated with the aged SSA factor is still primarily derived from sea spray (i.e., sea-salt  $SO_4^{2-}$ ), rather than from secondary formation via reactions between  $SO_2$  and SSA. This is supported by the molar ratio of  $SO_4^{2-}$  to  $SO_4^{2-}$  and  $SO_4^{2-}$  to  $SO_4^{2-}$  to  $SO_4^{2-}$  to  $SO_4^{2-}$  and  $SO_4^{2-}$  to  $SO_4^{2$ 

CS&B and AC&MS were major sources of fine-mode SO<sub>4</sub><sup>2-</sup>. Overall, AC&MS contributed more across all size bins (Fig. 12b). However, in winter with strong terrestrial transport, CS&B contributed more to droplet-mode SO<sub>4</sub><sup>2-</sup> (Fig. S12), and their contributions to total fine-mode SO<sub>4</sub><sup>2-</sup> were similar (49.7% for CS&B and 48.3% for AC&MS). As discussed above, shipping emission was another key contributor to fine-mode SO<sub>4</sub><sup>2-</sup> before IMO 2020 regulation, particularly for particles smaller than 0.32 µm. In summer, shipping emission even dominated the SO<sub>4</sub><sup>2-</sup> in this size range, with a contribution of 65.5% (Fig. S12). After the implementation of IMO 2020 regulation, the PMF-resolved contribution from shipping emission became negligible. However, this quantification may be uncertain and warrants further investigation.

Figure 12. Average fractional contributions of different sources to (a)  $NO_3^-$  and (b)  $SO_4^{2-}$  in size-segregated aerosols across all four seasons. Thick black lines represent the average measured  $NO_3^-$  concentrations in each size bin.

# 5 Conclusions and implications

595

The eastern China seas, as marginal seas adjacent to regions with intense human activities, experience atmospheric environment influenced by both anthropogenic pollution and natural emissions. The interactions between these factors significantly alter the physicochemical properties and environmental effects of aerosols in the marine boundary layer. This study conducted shipborne size-resolved aerosol sampling and analysis across four seasonal campaigns (2017–2020) in this region. Combined with size-resolved PMF analysis, we systematically examined the composition, size distribution, seasonal variations, major sources, and key processes of marine aerosols in this region.

Overall, aerosol mass concentration over the eastern China seas exhibits a trimodal distribution, with a coarse mode at 3.2–5.6 µm, a droplet mode at 0.56–1.0 µm, and a condensation mode at 0.18–0.32 µm. Sea salt dominates coarse-mode mass, except during intense dust transport events when dust aerosols prevail. In spring and winter, fine-mode aerosol mass exceeds coarse-mode mass, which differs from the characteristics of pristine open-ocean boundary layers where coarse-mode sea salt typically dominates the total mass concentration. This indicates the strong influence of continental anthropogenic pollution. While marine biogenic sources contribute to fine-mode aerosol mass through secondary formations and organic enrichment in SSA, the seasonal variations suggest they currently play a lesser role than continental transport. However, with future

reductions in anthropogenic emissions in China, whether marine biogenic sources will become a dominant contributor to fine-mode aerosols remains an important question for further study.

As continental secondary aerosols are transported over the ocean, fine-mode NH<sub>4</sub>NO<sub>3</sub> gradually decomposes, and fully aged aerosols lose their fine-mode NO<sub>3</sub><sup>-</sup>. Therefore, the presence of a prominent fine-mode NO<sub>3</sub><sup>-</sup> peak in marine aerosols signals fresh and strong continental pollution influence. Across all four cruises, the continental secondary aerosols in 50.9% of the samples are highly aged. The gaseous HNO<sub>3</sub> released from NH<sub>4</sub>NO<sub>3</sub> decomposition, along with NO<sub>x</sub> transported from land, undergo heterogeneous reactions with dust and sea salt, forming stable nitrate in coarse particles. On average, these reactions account for more than half (55.9%) of total NO<sub>3</sub><sup>-</sup> mass, with 43.2% associated with sea salt and 12.7% associated with dust. This process shifts NO<sub>3</sub><sup>-</sup> from fine mode to coarse mode, altering its transport distance and deposition pattern while also modifying the chemical composition, hygroscopicity, and ecological effects of dust and SSA. Modeling studies on aerosol deposition and its impact on surface phytoplankton should take this process into consideration.

Another key consequence of above heterogeneous reactions is chloride depletion in SSA, with an average depletion fraction of 27.7% across the four cruises and reaching 40.8% in summer. The chloride in SSA is transformed into reactive gases, which can produce chlorine radicals and influence ozone chemistry. Thus, anthropogenic NO<sub>x</sub> emissions influence not only photochemistry and ozone formation directly but also indirectly via SSA interactions and halogen activation. However, the spatial and temporal variations of this indirect effect and its response to future NO<sub>x</sub> emission changes require further investigation using modeling approaches.

Shipping emissions are another important source of aerosols over the eastern China seas, primarily contributing to particles smaller than 0.32 μm. Before the IMO 2020 regulation, shipping emissions accounted for ~20% of total SO<sub>4</sub><sup>2-</sup> mass, with contributions exceeding 50% to fine-mode SO<sub>4</sub><sup>2-</sup> in summer when continental influence was minimal. This significantly impacted regional air quality and CCN populations. Post-IMO 2020, the contribution of shipping emissions to aerosols likely dropped by an order of magnitude, suggesting the effectiveness of IMO 2020 regulation on controlling the ship-derived pollution. The concentrations of fine-mode V and Ni, two common tracers for shipping emission, decreased by 94.5% and 87.4%, respectively. As a result, shipping emissions are not the dominant source of these elements, making V and Ni no longer reliable tracers for ship emissions in bulk aerosol analysis. However, with size-resolved aerosol composition analysis, V in sub-0.32 μm particles can still serve as an indicator of ship emissions, although the changes in source profile must be considered in source apportionment studies.

This study provides a comprehensive characterization of major inorganic aerosol components over the eastern China seas and quantitatively examines key processes shaping aerosol composition and properties. These results improve our understanding of human impacts on marine aerosols and their environmental effects. The size-resolved PMF method used in this study effectively distinguishes aerosol sources across different size ranges, despite a limited sample count. Future studies integrating this approach with size-resolved organic composition analysis could enable deeper insights into aerosol sources

and processes. Additionally, this study offers unique observational data for constraining aerosol sources and aging processes in chemical transport models, which is essential for reducing model uncertainties.

# Code/Data availability

The EPA's PMF 5.0 software used in this study is publicly available from https://www.epa.gov/air-research/positive-matrix-factorization-model-environmental-data-analyses. Aerosol measurement data are available from the National Earth System Science Data Center, National Science & Technology Infrastructure of China (http://www.geodata.cn/data/datadetails.html?dataguid=24577094427168). The ship cruise trajectory information is available from the corresponding authors upon request (yingchen@fudan.edu.cn).

# **Author contribution**

YC and SZ conceived and designed the study. SZ collected aerosol samples. SZ and ZX performed the chemical analyses, 640 analyzed the data, and drafted the original manuscript with input from all other authors. YC reviewed the manuscript and finalized it.

# **Competing interests**

The authors declare that they have no conflict of interest.

### Acknowledgement

We gratefully acknowledge Honghai Zhang, Liang Zhao, and Xiangbin Ran for their leadership in conducting the cruise campaigns. We also appreciate the invaluable support of the scientific staffs and crew members aboard R/V *Dong Fang Hong III*, R/V *Dong Fang Hong III*, and R/V *Xiang Yang Hong XVIII* during the cruises. We acknowledge Haowen Li, Fnaghui Wang, Tianjiao Yang, and Yucheng Zhu for their assistance with aerosol sampling. We further acknowledge Xiping Ding, Ling Wen, and Qing Xu for their guidance on the ICP-MS instrument, as well as Tianfeng Guo for his help with OC/EC measurements.

# Financial support

Financial support has been received from the Natural Science Foundation of Shanghai (grant number 22ZR1403800), the National Key Research and Development Program of China (grant number 2016YFA0601304), and the open research cruise NORC2020-02 (R/V *Xiang Yang Hong XVIII*) supported by NSFC Shiptime Sharing Project (project number 41949902).

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
