# Peer review of "Anthropogenic air pollutants strongly interact with natural aerosols over the eastern China seas: key processes, size distributions, and seasonalities"

_EGUsphere, 2025_

## Author Comment (AC1)

**Responses to reviewers' comments**

We sincerely thank the two anonymous reviewers for providing very insightful evaluations and suggestions for the manuscript. Our point-by-point responses to reviewers' comments are listed in this file.

**Color Code:** Reviewers' comments, Authors' responses, Proposed changes in the manuscript

Line numbers before and inside the bracket refer to those in revised manuscript without and with track changes, respectively.

**1. Responses to Reviewer #1**

This manuscript investigates the seasonal chemical composition of particulate matter over the eastern China seas, based on shipborne measurements. A Size-resolved PMF analysis was employed to analyze the chemical composition data, and the effects of shipping emissions were discussed. Readers are frequently required to cross-reference multiple figures—often out of sequential order—to understand the presented arguments. For example, the discussion in Lines 299–307 necessitates referencing Figures 3 and 8 simultaneously, which disrupts the flow. A thorough reorganization of the manuscript structure and figure arrangement is strongly recommended to enhance readability.

Moreover, the key scientific conclusions are insufficiently clear. The title highlights the interaction between anthropogenic emissions and natural aerosols, but the main text lacks adequate discussion or supporting evidence on this topic. As currently presented, the work is more akin to a "measurement report" than a comprehensive "research article." Nevertheless, it may be considered for publication after the following major concerns are fully addressed:

We sincerely appreciate your insightful comments, which are very helpful in improving the quality of our manuscript. In the revised version, we have reorganized the structure of the *Discussion* section to enhance its logic flow and overall readability. The first subsection now presents a general discussion of the interactions between anthropogenic pollution and natural aerosols, and their effects on continental secondary aerosols and the size distribution of $NO_3^-$. To improve clarity, we have also included key chemical reaction equations in this section. Furthermore, we have added a more detailed analysis of the relationship between the $NO_3^-$ size distribution shift and relevant influencing factors, thereby providing stronger evidence for the proposed interaction mechanisms. Then how these interactions alter the properties of natural aerosols (i.e., dust and SSA) and their potential environmental implications are discussed in the following subsections.

Our point-by-point responses to your comments, along with the corresponding revisions in the manuscript, are provided below.

1.  Line 266-267: Is there any literature or observational evidence supporting the shift of nitrate from fine to coarse particles? I am also curious whether coagulation effects could contribute to the observed increase in coarse-mode nitrate concentrations. Since the coagulation of smaller particles may lead to the formation of larger ones—or smaller particles could be scavenged by larger particles—this process could also result in a shift of chemical species toward larger size fractions.

Thank you for your thoughtful comments. Yes, there are other previous studies have documented this thermodynamically driven shift in the size distribution of nitrate. For example, field measurements conducted in Shanghai (a coastal city in eastern China) revealed a significant redistribution of nitrate from the fine mode during non-dust periods to the coarse mode during dust events, and a laboratory experiment proves that the heterogeneous reactions between nitrogen-containing acidic gases and dust particles can account for this phenomenon (Wu et al., 2020). A chemical transport modeling study incorporating thermodynamic equilibrium modules (e.g., ISORROPIA-II) similarly showed a transition in nitrate size distribution in the presence of dust particles (Karydis et al., 2016). Specifically, the heterogeneous reaction between nitric acid and coarse dust particles forms coarse-mode nitrate, which

reduces the gas-phase nitric acid concentration and facilitates the evaporation of fine-mode ammonium nitrate. Another modeling study focused on the Asian dust outflow to the Pacific Ocean also highlighted the volatilization of ammonium nitrate and the resulting transfer of nitrate to dust particles, with the effect becoming more pronounced from coastal to remote ocean regions (Fairlie et al., 2010). Similar nitrate redistribution processes involving heterogeneous reactions with coarse sea-salt particles have also been observed in field studies (Savoie and Prospero, 1982; Pakkanen et al., 1996) and in modeling analyses (Chen et al., 2020; Myhre et al., 2006).

We agree that coagulation can lead to an increase in particle size. However, we believe it is not the primary cause of the observed increase in coarse-mode nitrate concentrations in our study. First, coagulation is highly inefficient for particles larger than 0.5 µm due to their low Brownian coagulation coefficients and number concentrations (Seinfeld and Pandis, 2016). The calculation by Jacobson et al. (1994) showed that coagulation has a negligible effect on the concentration and composition of supermicron particles. Second, if coagulation were the cause of nitrate redistribution to the coarse mode, we would expect to observe similar shifts for other species originally associated with fine particles, such as $NH_4^+$, $SO_4^{2-}$, and anthropogenic trace metal elements. However, such shifts were not observed in our data.

Therefore, we conclude that the heterogeneous uptake of nitrogen-containing gases by coarse dust or sea-salt particles—which inhibits the formation or enhances the evaporation of fine-mode ammonium nitrate—is responsible for the observed shift in nitrate size distribution.

2.    Section 3.3: The PMF model analyzed size-resolved chemical composition. However, the PMF factors were characterized by the size-integrated concentration. It is recommended to characterize PMF factors by both their size-integrated contributions and their mass size distributions. Additionally, diagnostic plots such as the Q/Qexp plot should be included in the SI to demonstrate the model's performance.

Thank you for your constructive suggestion. Our PMF analysis was conducted using the raw dataset, which includes the chemical composition of each size bin for every sample set. Both the size-integrated species contributions and the mass size distributions were considered when attributing PMF factors and are discussed in this section, with the corresponding figures provided. Specifically, Figure 7 presents the average normalized size distributions of aerosol concentrations associated with each PMF factor, while Figure S5 (previously Figure S3) displays the size distributions of factor contributions for individual sample sets. We have revised the language in the first paragraph of this section to improve clarity and minimize potential misunderstandings.

We have also added the $Q/Q_{exp}$ plots for the entire dataset and each chemical species to the Supplementary Materials (Figure S4a-b). The search range for the number of factors was extended from $5 – 8$ to $4 – 9$. The $Q/Q_{exp}$ curve for the overall dataset shows an elbow point at 7 factors. Furthermore, several key species (e.g., $Na^+$, $NH_4^+$, $Cl^-$, Cd) exhibit a marked decrease in $Q/Q_{exp}$ values when increasing the number of factors from 6 to 7. Based on these results and the physical interpretability of the factor profiles, we believe the 7-factor solution provides the most meaningful representation of the potential sources.

Additionally, we have included statistical metrics that quantify the agreement between measured and PMF-reconstructed concentrations (Figure S4c and Table S2). The PMF model effectively captures the concentration variations of all species, with $R^2 > 0.90$ for 29 out of 32 species, and $R^2 > 0.78$ for the remaining 3 species. These results further support the robustness of our PMF solution. A brief discussion of these results has been added to the manuscript accordingly.

**Lines 231-239 (241-251):** The SR-PMF model was run with factor numbers ranging from 4 to 9. Based on the trend of $Q/Q_{exp}$ with increasing factor numbers (Fig. S4a-b) and the physical interpretability of the solutions, we determine that the 7-factor solution provides the most reasonable results. The PMF-reconstructed concentrations show strong agreement with the measured values for all species, with $R^2 > 0.90$ for 29 out of 32 species and $R^2 > 0.78$ for the remaining 3 (Table S2 and Fig. S4c). The normalized contribution intensity of each PMF factor across different size ranges in each sample set is shown in Fig. S5. The average contributions to the size-integrated concentrations of different components (i.e., the chemical composition profiles) and the mean normalized size distributions of each factor are presented in Fig. 6 and Fig. 7, respectively. The attribution of each PMF factor to specific source or process is based on both its chemical composition profile and size distribution, as explained in detail below.

[Figure]

Figure S4. (a) Trend of the $Q/Q_{exp}$ value for the entire dataset as a function of the number of PMF factors. (b) Same as panel (a) but for each chemical species. (c) Time series of measured and PMF-reconstructed concentrations of five representative species. Each dot represents the concentration within a specific size bin of a sample set. Statistical comparisons between measured and PMF-reconstructed concentrations for other species are provided in Table S2.

Table S2. Statistical comparison between PMF-reconstructed and measured concentrations for each chemical species. The displayed metrices include the slope, intercept, and $R^2$ of the linear regression, and the normalized mean bias (NMB) between PMF-reconstructed and measured concentrations.

| | $Na^+$ | $NH_4^+$ | $K^+$ | $Mg^{2+}$ | $Ca^{2+}$ | $Cl^-$ | $NO_3^-$ | $SO_4^{2-}$ | $C_2O_4^{2-}$ | Al | Ba | Ca | Cd | Ce | Co | Cu |
|---|---|---|---|---|---|---|---|---|---|---|---|---|---|---|---|---|
| Slope | 0.982 | 0.801 | 0.717 | 0.897 | 0.732 | 0.998 | 0.731 | 0.746 | 0.740 | 0.743 | 0.940 | 0.700 | 0.703 | 0.808 | 0.836 | 0.614 |
| Intercept | 0.174 | 16.4 | 2.01 | −1.08 | 4.28 | −2.09 | 55.1 | 50.2 | 1.23 | 2.50 | −0.0341 | 8.25 | 3.28E−3 | 7.50E−3 | 6.04E−4 | 0.0625 |
| $R^2$ | 0.988 | 0.932 | 0.929 | 0.912 | 0.787 | 0.990 | 0.972 | 0.901 | 0.946 | 0.993 | 0.915 | 0.869 | 0.936 | 0.981 | 0.968 | 0.962 |
| MNB | 0.018 | 0.140 | 0.188 | 0.166 | 0.190 | 0.011 | 0.169 | 0.161 | 0.127 | 0.236 | 0.128 | 0.211 | 0.162 | 0.129 | 0.158 | 0.365 |

| | Eu | Fe | Ge | K | Mg | Mn | Mo | Na | Ni | Pb | Rb | Sb | Se | Sr | V | Zn |
|---|---|---|---|---|---|---|---|---|---|---|---|---|---|---|---|---|
| Slope | 0.702 | 0.893 | 0.709 | 0.715 | 0.817 | 0.798 | 0.534 | 0.977 | 0.519 | 0.760 | 0.703 | 0.636 | 0.716 | 1.009 | 0.762 | 0.629 |
| Intercept | 4.84E−5 | 1.54 | 1.20E−3 | 4.37 | 1.55 | 0.0989 | 0.0140 | 1.86 | 0.124 | 0.120 | 0.0124 | 0.0146 | 0.0219 | −6.47E−3 | 0.0764 | 0.605 |
| $R^2$ | 0.959 | 0.977 | 0.893 | 0.941 | 0.983 | 0.970 | 0.961 | 0.981 | 0.979 | 0.939 | 0.980 | 0.964 | 0.949 | 0.986 | 0.975 | 0.960 |
| NMB | 0.270 | 0.066 | 0.193 | 0.169 | 0.166 | 0.140 | 0.454 | 0.014 | 0.397 | 0.138 | 0.172 | 0.234 | 0.102 | 0.011 | 0.119 | 0.320 |

3.  Line 254-255: It is advisable to clearly indicate the co-transport events in Figure 8. Moreover, the specific dates of these events should be provided. Please clarify the method used to identify these events—was it based on abrupt changes in mass concentrations, backward trajectory analysis, or another approach?

Thank you for your suggestion. We have added brown-shaded bands in Figure 9 (previously Figure 8) to indicate the co-transport events, as shown below. The specific dates of these events have also been included in the revised text. These events were identified based on the time series of the contribution intensities of the PMF factors. Specifically, they correspond to concurrent strong increases in the contribution intensities of the dust factor and two continental pollution factors, i.e., primary anthropogenic emissions and CS&B. This has been explicitly noted in the revised manuscript. Additionally, during all three identified events, portions of the air mass backward trajectories originated from the Gobi Desert in Mongolia and passed through the densely populated regions of northern and eastern China (Figure S9), further supporting our interpretation.

**Lines 379-385 (393-398):** In this study, three strong co-transport events of dust and continental pollution aerosols (27 March – 28 March 2017, Sample 1; 14 April 2017, Sample 8 and 9; 22 October – 24 October 2020, Sample 10 and 11) were observed, as denoted by the events T1 – T3 in Fig. 9. These events are characterized by concurrent strong increases in the contribution intensity of three PMF factors: dust, primary anthropogenic emissions, and CS&B. Backward trajectory analyses (Fig. S9) indicate that the air masses associated with these events originated from the Gobi Desert in Mongolia and passed through the densely populated regions of northern and eastern China.

[Figure]

Figure 9. Temporal variations of the normalized contribution intensity of different sources and the mass concentrations of fine- and coarse-mode aerosols. The x-axis label represents the sample ID in chronological order within each cruise. T1 – T3 with brown shaded bands denote the three co-transport events of dust and continental pollution events.

[Figure]

Figure S9. 72-hour air mass backward trajectories during three co-transport events of dust and continental pollution aerosols: (a) Sample 1 from the spring campaign, (b) Samples 8-9 from the spring campaign, and (c) Samples 10-11 from the autumn campaign. The color of each trajectory point indicates the time before arrival at the sampling sites.

4.  Line 257-258: A discussion of the mass distribution of $NO_3^-$ and $SO_4^{2-}$ of each factor would be very helpful.

Thank you for your great suggestion. In the revised manuscript, we have reorganized the structure of the *Discussion* section. A new subsection (Section 4.5) has been added to specifically summarize the contributions of different sources to $NO_3^-$ and $SO_4^{2-}$ across different particle size ranges. The size-resolved source contributions to all-season averages are now presented in Figure 12 (previously Figure 9), while the seasonal results have been moved to the Supplementary Materials as Figures S11 and S12 (previously Figures 10 and 11).

**Lines 554-580 (639-665):**

**4.5 Summary of $NO_3^-$ and $SO_4^{2-}$ sources across particle sizes**

This section summarizes the contributions of different sources to $NO_3^-$ and $SO_4^{2-}$ across different particle sizes. Consistent with the previous discussion, dust and aged sea salt aerosol (SSA) are the two dominant contributors to coarse-mode $NO_3^-$ over the eastern China seas. This reflects the importance of heterogeneous reactions between nitrogen-containing acidic gases and dust or SSA particles. On average across all four seasons, these two pathways together accounted for more than 85% of $NO_3^-$ in all size bins larger than 1.8 µm (Fig. 12a). Specifically, dust contributed 17% to 39% to $NO_3^-$ in different size bins of coarse particles, with an overall contribution of 20.5%. Aged SSA contributed a higher proportion (72.0%), ranging from 52% to 75% in coarse particles and exceeding 70% in the 1.8–10 µm range. This strong contribution of aged SSA to coarse-mode $NO_3^-$ was consistent across all seasons, and particularly prominent in summer, when aged SSA accounted for 95.4% of total coarse-mode $NO_3^-$ (Fig. S11). Continental secondary formation was the dominant source of fine-mode $NO_3^-$, accounting for over 90% of the all-season average $NO_3^-$ concentration in particles smaller than 1.0 µm (Fig. 12a). This pattern held for the seasonal averages in spring, autumn, and winter (Fig. S11). While in summer, due to the extremely low concentrations and high measurement uncertainties, the PMF results for submicron $NO_3^-$ may not be statistically robust.

Sea salt (fresh SSA + aged SSA) contributed 56.8% of all-season average coarse-mode $SO_4^{2-}$ concentration, and this relative contribution peaked at 5.6 – 10 µm (73.1%, Fig. 12b). In autumn, when SSA concentrations were highest, the contribution to coarse-mode $SO_4^{2-}$ reached 70.9%. It is worth noting that $SO_4^{2-}$ associated with the aged SSA factor is still primarily derived from sea spray (i.e., sea-salt $SO_4^{2-}$), rather than from secondary formation via reactions between $SO_2$ and SSA. This is supported by the molar ratio of $SO_4^{2-}$ to $Na^+$ from these two SSA factors, which is 0.052, closely matching the seawater ratio of 0.058. In addition to SSA, the two secondary sources CS&B and AC&MS also made substantial contributions to coarse-mode $SO_4^{2-}$, particularly in spring and winter, when their combined contributions were 48.5% and 50.0%, respectively.

CS&B and AC&MS were major sources of fine-mode $SO_4^{2-}$. Overall, AC&MS contributed more across all size bins (Fig. 12b). However, in winter with strong terrestrial transport, CS&B contributed more to droplet-mode $SO_4^{2-}$ (Fig. S12), and their contributions to total fine-mode $SO_4^{2-}$ were similar (49.7% for CS&B and 48.3% for AC&MS). As discussed above, shipping emission was another key contributor to fine-mode $SO_4^{2-}$ before IMO 2020 regulation, particularly for particles smaller than 0.32 µm. In summer, shipping emission even dominated the $SO_4^{2-}$ in this size range, with a contribution of 65.5% (Fig. S12). After the implementation of IMO 2020 regulation, the PMF-resolved contribution from shipping emission became negligible. However, this quantification may be uncertain and warrants further investigation.

5. Line 258-259: Please consider quantifying and presenting the mass contributions of different PMF factors to $NO_3^-$. A bar chart or tabulated data could greatly improve clarity.

Thank you for your suggestion. We have presented the contributions of different PMF factors to the $NO_3^-$ across different size bins in Figure 12a (previously Figure 9a). The overall contributions to the size-integrated $NO_3^-$ concentration were previously shown in Figure 6 and Figure S6 (formerly Figure S4) as bar plots, but the numerical values were not provided. To address this, we have now added Tables S3 – S5 in the Supplementary Materials to explicitly present these values.

6. Line 280-283: Can the author provide a rough estimate of the mass contribution of the $NaNO_3$ formation pathway to coarse-mode nitrate? Since the conclusion is primarily based on size distribution data, it would be more convincing to support it with additional evidence or quantitative estimation.

The contributions of heterogenous reactions with dust and sea salt to the coarse-mode nitrate have been quantified by the SR-PMF analysis, with the results presented in Figure 12a (previously Figure 9a) and Figure S11 (previously Figure 10) and discussed in detail in the newly added Section 4.5. According to our analysis, heterogenous reactions with dust and sea salt contribute 20.5% and 72.0% of coarse-mode nitrate, respectively. These values are also included in the newly added Table S5.

**Lines 559-561 (644-646):** Specifically, dust contributed 17% to 39% to $NO_3^-$ in different size bins of coarse particles, with an overall contribution of 20.5%. Aged SSA contributed a higher proportion (72.0%), ranging from 52% to 75% in coarse particles and exceeding 70% in the 1.8–10 µm range.

7.  Line 294-296: Please specify which figure or table supports this conclusion. It is important to ensure that every claim is clearly linked to the corresponding data presentation.

Thank you for pointing this out. We have added appropriate references to the supporting figure and table.

**Lines 350-352 (363-365):** As a result, AC&MS contributes significantly to the observed Se concentrations in marine aerosols (Fig. 6), explaining why Se concentrations do not show the typical drastic summer decline observed for other terrestrial trace elements (Table S1).

8.  Line 301-302: It is difficult to identify the "strong impact" samples in Figure 3. Consider providing a separate figure showing the size distribution of $NO_3^-$ during the periods classified as having strong anthropogenic influence. Additionally, please clarify the criteria used to define these periods.

Thank you for your great suggestions. We have added a new figure to the Supplementary Materials (Figure S7) to illustrate the distinct size distributions of $NO_3^-$ under conditions with and without strong influence from continental anthropogenic pollution. This "strong impact" was identified based on the following criteria:

(a) A prominent accumulation-mode or droplet-mode peak in the $NO_3^-$ size distribution, defined as

either $\frac{[NO_3^-]_{0.18-0.32}}{\frac{1}{2}\times([NO_3^-]_{0.10-0.18}+[NO_3^-]_{0.32-0.56})} > 1$ or $\frac{[NO_3^-]_{0.56-1.0}}{\frac{1}{2}\times([NO_3^-]_{0.32-0.56}+[NO_3^-]_{1.0-1.8})} > 1$.

(b) More than 15% of the $NO_3^-$ concentration is distributed in $PM_{1.8}$.

(c) CS&B contributes more than 15% of the total $NO_3^-$ concentration.

These criteria are now clearly described in the revised manuscript. Additionally, we would like to note a correction regarding the number of samples previously identified as being strongly influenced by continental pollution. The correct number is 26, not 27, and we have amended this in the revised version accordingly.

**Lines 313-318 (326-331):** Across all four cruise campaigns, 49.1% (26/53) of the aerosol samples were identified as strongly influenced by this process, based on the following three criteria: (1) there is a prominent accumulation-mode or droplet-mode peak in the $NO_3^-$ size distribution, defined as $\frac{[NO_3^-]_{0.18-0.32}}{\frac{1}{2}\times([NO_3^-]_{0.10-0.18}+[NO_3^-]_{0.32-0.56})} > 1$ or $\frac{[NO_3^-]_{0.56-1.0}}{\frac{1}{2}\times([NO_3^-]_{0.32-0.56}+[NO_3^-]_{1.0-1.8})} > 1$; (2) more than 15% of the $NO_3^-$ concentration is distributed in particles with a diameter smaller than 1.8 μm ($PM_{1.8}$); and (3) the CS&B factor contributes more than 15% of the total $NO_3^-$ concentration. $NO_3^-$ in these samples exhibits bimodal or trimodal size distributions (Fig. S7).

[Figure]

Figure S7. Two distinct types of $NO_3^-$ size distribution: (a) bimodal or trimodal distribution, indicating a strong influence from continental anthropogenic pollution, with relatively fresh pollution aerosols; (b) unimodal distribution, suggesting a relatively weak direct influence from continental pollution, with highly aged pollution aerosols.

9. Line 303-304: Please provide some discussion of air masses, such as wind filed in different seasons or backward trajectory analysis.

Thanks for your great suggestion. We have added a supplementary figure (Figure S2) presenting the 72-hour backward trajectories arriving at the sampling sites (i.e., ship locations) during the four cruise campaigns. This figure clearly illustrates that most air mass trajectories in summer originate from oceanic regions, whereas a substantially greater proportion in the other seasons originate from continental regions. This seasonal contrast aligns well with the established understanding of the East Asian monsoon's influence on air mass transport patterns.

**Lines 160-163 (167-170):** This seasonal variation aligns with the influence of the East Asian monsoon on air mass transport patterns, i.e., southeasterly winds from the cleaner oceanic environment dominate in summer, while northwesterly winds bring more pollution from the land in winter (Zhou et al., 2023). This pattern is also clearly supported by air mass backward trajectory analysis (Fig. S2).

**Lines 319-321 (332-334):** In summer, as air masses primarily originate from the southeastern ocean (Fig. S2), the direct influence of terrestrial transport and thus the overall contribution from CS&B is significantly weaker (Fig. 9).

[Figure]

Figure S2. 72-hour air mass backward trajectories arriving at the sampling sites during the four cruise campaigns. The color of each trajectory point indicates the time before arrival. Only trajectories with arrival times falling within the MOUDI sampling periods are shown.

10. Line 305-306: Please specify which sample(s) in Figure 3 were influenced by sea fog conditions. A note or label in the figure would help readers easily identify these cases.

Thank you for your suggestion. In response to Comment #8, we have added a new figure (Figure S7) to separately display the two distinct types of $NO_3^-$ size distributions, and we now reference this figure in the revised sentence. Accordingly, we have added the sea fog label to Figure S7 instead of Figure 3.

[Figure]

Figure S7. Two distinct types of $NO_3^-$ size distribution: (a) bimodal or trimodal distribution, indicating a strong influence from continental anthropogenic pollution, with relatively fresh pollution aerosols; (b) unimodal distribution, suggesting a relatively weak direct influence from continental pollution, with highly aged pollution aerosols.

11. Line 308-310: The $R^2$ value in Figure S5 represents the fit of the linear regression model for $PM_{1.8}$ mass. However, it does not necessarily imply that CS&B and AC&MS explain 70% of the variance in $PM_{1.8}$. Notably, the regression slope is about 0.5, suggesting that CS&B and AC&MS jointly account for only ~50% of $PM_{1.8}$ mass. Please revise this statement to accurately reflect the regression results.

Thank you for pointing out this issue. We agree that the $R^2$ value does not necessarily imply the explained variance or the mass fraction contributed to $PM_{1.8}$ concentration. Instead, the regression slope provides a better measure of relative contribution. In our analysis, the slope for the combined CS&B and AC&MS sources is 0.366. Including additional factors increases the slope only slightly to 0.384. However, since organic aerosols were not included in our PMF analysis due to the lack of data in the autumn and winter campaigns, the actual contribution is likely much higher. Thus, the current slope represents only a lower bound.

Previous studies have shown that organics can account for approximately half of the fine particle mass in eastern China, with the majority originating from combustion-related emissions and secondary formation processes (Daellenbach et al., 2024; Huang et al., 2014). Therefore, it is reasonable to infer that CS&B and AC&MS are the dominant sources of $PM_{1.8}$, particularly in spring, autumn, and winter when continental pollution outflows are more prevalent. We have revised the figures and discussion accordingly.

**Lines 360-369 (373-382):** Summing the contributions of each PMF factor to the total concentrations of the chemical species included in the SR-PMF analysis, we find that the combined contribution from CS&B and AC&MS factors is strongly correlated with the $PM_{1.8}$ mass concentration ($R^2 = 0.678$; Fig. S8a). It is important to note that organics were not included in the SR-PMF analysis due to the lack of measurement during the autumn and winter campaigns. Consequently, the PMF-reconstructed concentrations are significantly lower than the observed $PM_{1.8}$ mass concentrations, with a linear regression slope of 0.366 (Fig. S8a). Including other PMF factors only slightly improves the regression slope, from 0.366 to 0.384 (Fig. S8). Organic aerosols are known to be abundant in fine particles over eastern China, with concentrations comparable to those of inorganic components, and they predominantly originate from combustion-related sources and secondary formation processes (Daellenbach et al., 2024; Huang et al., 2014). Therefore, it is likely that CS&B and AC&MS represent the primary sources of fine particles in our study, especially during spring, autumn, and winter.

[Figure]

Figure S8. Correlations between $PM_{1.8}$ mass concentrations and the total concentrations of PMF species in sub-1.8 μm particles contributed by (a) CS&B and AC&MS, and (b) all PMF factors. Here PMF species correspond to all chemical species included in PMF analysis but excluding duplicate species (e.g., $Na^+$ is excluded because elemental Na is also included). Different colors represent different seasons.

12. Line 310-311: Please clarify what is meant by "explained variance." If referring to statistical variance explained by regression, it should be supported with appropriate statistical metrics.

In light of your previous comment, we have revised our interpretation and removed the discussion related to explained variance. Please refer to our earlier response for further details.

13. 10: I It is unexpected that the aged sea salt factor contributes notably to $NO_3^-$ in the fine size range (0.1–0.32 μm). The manuscript previously suggested negligible fine-mode $NO_3^-$ for highly aged aerosols. Please provide further explanation or hypothesis for this observation.

Thank you for your question. The primary reason is the high uncertainty caused by the extremely low concentrations. As indicated by the thick black line in Figure S11 (previously Figure 10), the concentration of $NO_3^-$ in particles smaller than 0.56 μm during summer is very low ($< 0.025$ μg m$^{-3}$). Consequently, both the measurement uncertainty and the resulting uncertainty in the PMF analysis are

relatively large. In this context, a single sample showing a relatively high contribution from the aged sea salt factor—even if the absolute concentration is low—can disproportionately influence the seasonal average contribution attributed to that factor. However, this uncertainty associated with extremely low concentration samples has minimal impact on the overall source profile results when considering multi-season averages (Figure 12a). To clarify this, we have added a note to the figure caption.

**In the caption of Figure S11:** Note that source contributions in size bins with very low $NO_3^-$ concentrations (e.g., $D_p < 0.56$ μm in summer) are subject to considerable uncertainty and may not be statistically robust; however, this has minimal impact on the overall source profile.

14. Line 341-342: How was the "3.4 times" increase calculated? Was this based on empirical data or cited from previous literature? Please clarify the basis and, if calculated, include the equation or data used.

This statement is based on the PMF-derived contributions of aged SSA and dust to $NO_3^-$ concentrations, as shown in Figure 6 and Figure S6 (previously Figure S4). We have relocated this sentence to follow the discussion of the contribution of aged SSA to $NO_3^-$ and have added specific numerical values to enhance clarity.

**Lines 423-427 (497-503):** As noted in Section 4.1, the heterogeneous reactions between SSA and nitrogen-containing acidic gases shift $NO_3^-$ from smaller to larger particle sizes (Chen et al., 2020), greatly affecting its atmospheric lifetime and transport distance. Based on the SR-PMF results, the average concentration of $NO_3^-$ associated with aged SSA was 1.47 μg m⁻³, accounting for 43.2% of the total $NO_3^-$ concentration (Figs. 6 and S6, Table S3). This contribution is 3.4 times greater than that with dust (i.e., represented by dust factor, 0.43 μg m⁻³), indicating that SSA plays a more significant role in the size redistribution of $NO_3^-$.

15. Line 342-345: Same concern as in Comment 11: The use of R² and the slope should be carefully interpreted. Please rephrase this sentence to ensure the conclusion aligns with the regression outcome

Thank you for your comment. The average concentration of all species integrated from both fresh and aged SSA factors accounts for 57.2% of the average PM$_{>1.8}$ concentration. This sufficiently supports our statement that SSA typically dominates the coarse aerosol mass. To avoid potential misinterpretation, we have therefore removed the discussion of the regression results in the revised manuscript.

**Lines 428-431 (508-514):** Summing the concentration of all species contributed by fresh and aged SSA factors, their combined average concentration accounts for 57.2% of the average PM$_{>1.8}$ mass. This proportion would likely be even higher if components not included in the SR-PMF analysis were considered, indicating SSA generally dominates the coarse-mode aerosol concentration.

16. Line 345-346: Please provide data corresponding to strong dust transport events. Clarify which specific samples were collected during these events and consider highlighting them in relevant figures or tables.

Thank you for your great suggestion. We have clarified the sampling period (22 October – 24 October 2020), and this is also highlighted as the event T3 in Figure 9 (previously Figure 8).

Regarding the detailed data, PMF analysis directly quantifies the contribution of dust sources to the chemical species included in the model. To estimate the dust aerosol mass, we further converted the concentrations of trace elements into their common oxide forms using the dust equation from Liu et al. (2022b):

$$dust = [1.89Al + 2.14Si + 1.40Ca + 1.36Fe + 1.67Ti + 1.20K + 1.66Mg + 1.35Na] \times CF$$

Here, the concentrations of all elements correspond to the dust-contributed concentrations derived from the PMF analysis. Since Si and Ti were not directly measured in this study, their concentrations were estimated using previously reported Si/Al and Ti/Al ratios for East Asian dust, which are 3.54 and 0.06, respectively (Liu et al., 2022b). The correction factor (CF) applied for East Asian dust is 1.11. After this reconstruction, the PMF-resolved concentration of $NO_3^-$ associated with dust was also added.

Combing the PMF results with the dust equation, we estimated the coarse-mode dust aerosol concentration during this period to be 32.1 µg m⁻³, accounting for 65.9% of the $PM_{>1.8}$ mass concentration.

**Lines 433-435 (516-518):** By combining the PMF results with a dust equation that convert trace elements to their common oxide forms (Liu et al., 2022b), the coarse-mode dust aerosol concentration was estimated at 32.1 µg m⁻³, accounting for 65.9% of the $PM_{>1.8}$ concentration during this period.

17.  Line 351-354: Please elaborate on how the faster reaction rate of aged sea salt results in a smaller median particle diameter. Could you also provide a rough estimation or reference regarding the reaction rate differences between aged and fresh sea salt?

Thanks for your question. First, we would like to clarify an important point regarding SSA aging. In the real atmosphere, SSA particles are typically only partially aged. An SSA particle can be viewed as an internal mixture of unreacted sea salt components and those in which Cl⁻ has been replaced by $NO_3^-$. In this study, these two parts are represented by the fresh sea salt and aged sea salt PMF factors, respectively. In other words, these two PMF factors should not be interpreted as representing an external mixture of fully fresh and fully aged SSA particles. We have added a clarification in the revised manuscript to reflect this point. Following this definition, it is not "the faster reaction rate of aged sea salt", but the surface area dependence of the reaction rate between fresh SSA and acidic gases resulting in the smaller mass median diameter of aged SSA factor (i.e., the reacted portion) compared to the fresh SSA.

The initial step in the heterogeneous reaction between SSA and acidic gases is the uptake of gas molecules onto the particle surface. This uptake rate is proportional to the available aerosol surface area and often governs the overall reaction rate (Rossi, 2003). Assuming a constant density for fresh SSA across different sizes, the aerosol surface area concentration is approximately proportional to the mass concentration divided by particle size. As shown in the newly added Figure S10, the normalized size distribution of [Fresh sea salt]/$D_p$ (a proxy for the surface area of fresh sea salt, where [Fresh sea salt] denotes the PMF-resolved source intensity of fresh sea salt) closely resembles the size distribution of the aged sea salt factor. This supports our interpretation. Similarly, previous studies have reported

slightly smaller median sizes for $NO_3^-$ associated with aged SSA compared to $Na^+$, and attributed this to the surface-area dependence of the heterogeneous reaction rate (Zhuang et al., 1999). We have added further discussion on this point in the revised manuscript.

**Lines 440-445 (523-528):** This difference is linked to the surface-area dependence of heterogeneous reaction rates between SSA particles and acidic gases, as gas uptake onto aerosols is proportional to aerosol surface area and often limits the overall reaction rate (Zhuang et al., 1999; Rossi, 2003). As shown in Fig. S10, the normalized size distribution of [Fresh sea salt]/$D_p$ (a proxy for the surface area of fresh sea salt) closely resembles the size distribution of the aged sea salt factor, supporting the surface-area-limited nature of the heterogeneous reaction mechanism.

[Figure]

Figure S10. Normalized size distributions of average concentrations from fresh sea salt and aged sea salt PMF factors, along with the ratio of fresh sea salt to particle diameter ([Fresh sea salt]/$D_p$), which serves as a proxy for the surface area of fresh sea salt. The abnormally high value of [Fresh sea salt]/$D_p$ in the smallest size bin is likely due to large measurement and PMF uncertainties associated with the extremely low sea salt concentrations in that range.

18. Line 356-357: Clarify how the chloride depletion ratio was determined. Was it calculated from molar ratios of $Na^+$ and $Cl^-$ in the measurement data, or derived from another method?

The chloride depletion ratio was calculated based on the molar ratios of $Na^+$ and $Cl^-$ in the measurement data. We assume that $Na^+$ in supermicron particles is a conservative tracer for sea salt, originating solely from sea spray and unaffected by atmospheric chemical reactions. Then, the chloride depletion ratio ($Cl_{dep}$) is calculated by:

$$Cl_{dep} = 1 - \frac{[Cl^-]}{[Na^+] \times \left(\frac{[Cl^-]}{[Na^+]}\right)_{SSA}}$$

Here, $[Cl^-]$ and $[Na^+]$ represent the measured molar concentrations of $Cl^-$ and $Na^+$, respectively. $\left(\frac{[Cl^-]}{[Na^+]}\right)_{SSA}$ equals 1.16, which is the theoretical molar ration of $Cl^-$ to $Na^+$ in fresh sea salt aerosols—the same as the molar ratio in seawater. We have added this explanation to the revised manuscript.

**Lines 449-452 (533-535):** Here the chloride depletion ratio ($Cl_{dep}$) was calculated based on the measured molar concentrations of $Cl^-$ and $Na^+$: $Cl_{dep} = 1 - [Cl^-]/(1.16 \times [Na^+])$, where 1.16 is the abovementioned theoretical molar ration between $Cl^-$ and $Na^+$ in fresh sea salt aerosols, consistent with seawater composition.

19. Line 376-378: Please present the seasonal variation in the size distribution of $Ca^{2+}$ in either figure or tabular format to support the corresponding discussion.

Thank you for raising this point. The seasonal variation in the size distribution of $Ca^{2+}$ is presented in Figure S3 (previously Figure S2). However, we inadvertently omitted a reference to it in the previous manuscript. We have now added a citation to Figure S3 in the relevant sentence.

[Figure]

Figure S3. Size distributions of (a) $K^+$, (b) $Mg^{2+}$, (c) $Ca^{2+}$, and (d) $C_2O_4^{2-}$ during four cruises. Each number represents a sample set. The thick black line with gray area below represents the average size distribution for each cruise.

20. Line 387-394: Although the text discusses the seasonal variation of V and Ni, the corresponding data are not presented. Please include a figure or table showing the concentration and size distribution of V and Ni in different seasons.

We apologize for missing the references to the figures and tables. The seasonal size distributions of V are presented in Figure 5. Here we have added Ni to this figure and added the reference to it in the relevant sentence. Additionally, the values of fine- and coarse-mode concentrations across different seasons are provided in Table S1, and we have added the corresponding reference as well.

**Lines 488-490 (573-575):** In contrast, during the autumn and winter cruises, these values dropped to 0.64 and 1.11 ng m$^{-3}$, representing decreases of 94.5% and 87.4%, respectively (Fig. 5 and Table S1).

[Figure]

Figure 5. Size distributions of (a) Al, (b) Cd, (c) Se, (d) V, and (e) Ni during four cruises.

21. Line 464: Please provide additional evidence to support the claim of fine-mode $NH_4NO_3$ decomposition, such as thermodynamic analysis, ambient temperature data, or references to previous studies

Thank you for your great suggestion. In response, we have added a detailed analysis of the evolution of $NO_3^-$ size distribution in relation to air mass transport history, air temperature, and SSA concentrations. The results clearly show that the concentration ratio between fine-mode and coarse-mode $NO_3^-$ gradually decreases as air masses spend more time over the ocean (i.e., as continental aerosols become more aged). Additionally, elevated temperatures and higher SSA concentrations appear to promote this fine-to-coarse transition (Figure 8), supporting the proposed transformation mechanism involving the thermodynamic decomposition of fine-mode $NH_4NO_3$ and heterogeneous reactions between nitrogen-containing acidic gases with coarse-mode SSA (as well as dust aerosols during strong dust transport events).

The following two paragraphs, extracted from the revised manuscript, provide a detailed explanation of this analysis. Further discussion of the thermodynamically driven $NO_3^-$ redistribution mechanism, along with references to previous studies, can be found in the same subsection (Section 4.1.1).

**Lines 289-308 (302-321):**

This $NO_3^-$ redistribution mechanism, along with the aging of continental secondary aerosols, is supported by our observations. As shown in Fig. 8, the concentration ratio of fine-mode to coarse-mode $NO_3^-$ decreases significantly with increasing time that the 72-hour air mass backward trajectories spent over the ocean. Specifically, when air masses had travelled over the ocean for less than 24 hours—i.e., when continental aerosols were relatively fresh—the median fine-to-coarse ratio of $NO_3^-$ was 2.23 (Fig. 8b), indicating that in more than half of the samples, over two-thirds of the $NO_3^-$ mass remained in fine particles. In contrast, when the time over ocean exceeded 48 hours (i.e., the continental aerosols were highly aged), the fine-to-coarse ratio fell below 1 in 92.6% of the samples (even below 0.15 in 59% of samples). Although an extreme outlier appears in the top-right corner of Fig. 8a (time over ocean = 71.8 hours, fine-to-coarse ratio = 4.0; Sample 5 from the winter campaign), this anomaly is explainable because the research vessel passed through the Zhoushan archipelago which hosts the world largest port locates and has a relatively high population density, and was likely strongly influenced by anthropogenic pollution.

Beyond the time over ocean, the fine-to-coarse ratio of $NO_3^-$ is also strongly influenced by SSA abundance (represented by $Na^+$ concentration here). As shown in Fig. 8a, at a given time over ocean level, samples with higher $Na^+$ concentrations generally exhibited lower fine-to-coarse $NO_3^-$ ratios. In addition, during the most intense dust transport event in this study (22 October – 24 October 2020, marked by the two filled circles in Fig. 8a), the observed fine-to-coarse ratios were significantly lower than typical values at similar levels of time over ocean and $Na^+$ concentrations. These results further support the role of SSA and dust in facilitating the transformation of $NO_3^-$ from fine to coarse mode via heterogeneous reactions. Furthermore, air temperature also appears to influence $NO_3^-$ partitioning. As shown in Fig. 8a, higher temperatures are typically associated with lower fine-to-coarse ratios. When the average air temperature along the backward trajectory exceeded 15 °C, all samples exhibited fine-to-coarse ratios below 1, except for two cases with very low $Na^+$ concentrations.

[Figure]

Figure 8. (a) Scattering plot between the concentration ratio of fine-mode to coarse-mode $NO_3^-$ (Fine-to-coarse ratio$_{[NO_3^-]}$) and the time that 72-hour air mass backward trajectories spent over the ocean (Time over ocean). The color and size of the scatter points indicate the average air temperature along the trajectory (Temperature$_{traj}$) and the size-integrated $Na^+$ concentration, respectively. The two filled circles highlight cases observed during the strongest dust event (22 October – 24 October 2020). (b) Boxcharts of the  Fine-to-coarse ratio$_{[NO_3^-]}$  grouped by different ranges of Time over ocean. The boxes represent the interquartile range (25th to 75th percentiles), the horizontal lines indicate the median, and the whiskers represent the highest and lowest values within median ± 1.5 interquartile range. Red circles denote the mean values, and stars indicate statistical outliers.

22.  Line 474-477: Bromide and halogen radicals are mentioned in the conclusion, yet they are not discussed in the main body of the text. Please either integrate the relevant discussion into the manuscript or remove this sentence from the conclusion to avoid confusion.

Thank you for your insightful suggestion. We agree that discussing the potential roles of bromide and halogen radicals in the conclusion may not be appropriate, given the lack of detailed analysis in our study. In response, we have moved this discussion to Section 4.3.1 and integrated it with the discussion of the potential environmental implications of SSA aging.

**Lines 456-460 (540-545):** HCl and $ClNO_2$ released from SSA aging serves as an important source of reactive chlorine in the atmosphere, which has been shown to enhance ozone formation in polluted regions (Li et al., 2021; Wang et al., 2020). In addition to chloride, bromide in SSA can undergo similar processes, releasing reactive bromine gases and further generating bromine-containing radicals (Parrella et al., 2012; Sander et al., 2003). Collectively, these reactive halogen gases from sea salt aging over coastal seas may influence ozone pollution in eastern China, though the extent of this influence requires further investigation.

**2. Responses to Reviewer #2**

This study investigates the chemical composition, size distribution, and source contributions of aerosols over the eastern China seas based on four seasonal cruise campaigns. Using size-resolved sampling and PMF analysis, the authors identify key anthropogenic and natural sources. The work highlights important processes such as NO3 redistribution and Cl depletion, and discusses their implications for regional air quality and deposition.

While the study presents a comprehensive dataset and detailed analyses, the connection between the results and discussion sections is somewhat fragmented. A few conclusions are not clearly supported by the data. The four parts of the discussion read more like standalone topics rather than integrated components of one topic.

I recommend addressing the following comments to improve the clarity of the manuscript.

We sincerely appreciate your insightful comments, which are very helpful in improving the quality of our study and manuscript. In the revised version, we have reorganized the structure of the *Discussion* section to enhance its coherence and clarity. Specifically, we now begin the section with an overarching discussion on the interactions between anthropogenic pollution and natural aerosols, and their influence on continental secondary aerosols and $NO_3^-$ size distribution. To improve clarity, we have included key chemical reaction equations related to these interactions. We also added a more detailed analysis of the relationship between the shift in $NO_3^-$ size distribution and the relevant influencing factors, providing additional evidence to support our interpretation. Subsequently, we discuss how these interactions modify the properties of natural aerosols (i.e., dust and SSA), along with their potential environmental implications, in the following subsections.

Our detailed responses to each specific comment and the corresponding revisions are presented below.

1. The autumn cruise sampling was conducted during the COVID pandemic. Have the authors considered the potential influence of COVID-related emission changes? How might this affect the observations in this study?

Thank you for pointing out this issue. As reported in previous studies (Doumbia et al., 2021; Zheng et al., 2021), substantial COVID-induced reductions in major air pollutant emissions from China (i.e., >10%) occurred primarily in February and March 2020. Emissions rebounded quickly thereafter and returned to levels comparable to 2019 in the second half of 2020, with relative differences less than 5%. A similar pattern was observed in $PM_{2.5}$ concentrations in northern China, which showed significant decreases in February and March 2020, followed by a return to typical concentrations in the subsequent months (Zheng et al., 2021). As shown in the newly added Figure S2, some of the air mass backward trajectories during the second half of the autumn campaign also passed over Korea. To evaluate potential COVID-related impacts, we examined the percentage changes in air pollutant emissions across different source sectors in Korea based on the CONFORM (COvid-19 adjustmeNt Factors fOR eMissions) dataset (https://eccad.sedoo.fr/#/metadata/545) (Doumbia et al., 2021). As illustrated in Figure C1, COVID-induced emission changes varied throughout the year. Nonetheless, during the second half of the autumn campaign, emission reductions from all sectors remained within 10%.

In summary, while we recognize that changes in human activity associated with the COVID pandemic could potentially affect aerosol loading over nearby oceanic regions, we believe that the impact on our observations was minimal. A brief discussion on this point has been added to the revised manuscript.

[Figure]

Figure C1. Relative changes in anthropogenic air pollutant emissions from different source sectors in Korea during 2020 due to the COVID pandemic. The light orange shaded area indicates the period during which the autumn campaign was conducted.

**Lines 370-375 (383-388):** The autumn campaign was conducted after the outbreak of COVID-19 pandemic, during which anthropogenic emissions may have been reduced by control measures such as lockdowns. However, the most stringent lockdowns and associated emission reductions in East Asia occurred in the first half of 2020. By the second half of the year, emissions had largely returned to 2019 levels, with relative differences below 10% (Doumbia et al., 2021; Zheng et al., 2021). Therefore, although some influence from COVID-related measures cannot be ruled out, they are not expected to have significantly affected our observations.

2.    Additionally, for the autumn cruise, it appears that samples 9, 10, and 11 were collected at the same location, which is very close to land. How could this affect the results? Do these three samples reflect more emissions from port activities (e.g., shipping emissions, primary anthropogenic sources)? For example, in Figure 3, sample 10 from the autumn cruise shows much higher SO4, NO3, and NH4 concentrations and lower Na and Cl concentrations than other autumn samples. Samples 9–11 also appear to show different levels of C2O4 and trace metals.

Yes, these three samples were collected at the same location. During this period, the sea surface experienced strong winds and rough wave conditions, so the research vessel temporarily anchored along the coast of Zhejiang Province to shelter from the adverse weather. The anchoring site was located outside of any port area and was therefore not directly affected by intense port activities.

We acknowledge that, due to the generally higher ship density in coastal regions and the influence of anthropogenic emissions from nearby land, these samples were relatively more impacted by local

anthropogenic pollution. However, long-range transport played a more significant role during this period.

Samples 10-11 correspond to the third co-transport events of dust and continental pollution aerosols. During this event, air masses originated from the Gobi Desert in Mongolia and subsequently passed over northern and eastern China (see Figure C2b; additional details are provided in our response to Comment #6). Driven by strong northerly winds, anthropogenic pollutants from northern and eastern China were transported to the southeastern coast ahead of dust from more distant regions. This transport pattern is consistent with our observations: the peak concentrations of anthropogenic pollutants and the PMF-derived source contributions from primary anthropogenic emissions and CS&B occurred in Sample 10, while the peak dust concentration and corresponding PMF source intensity were observed later in Sample 11 (Figure 9, previously Figure 8). In contrast, the air masses associated with Sample 9 spent a longer duration over the ocean (Figure C2a). Correspondingly, the observed concentrations of anthropogenic pollutants in Sample 9 were substantially lower than in Sample 10. If the aerosols were primarily driven by local pollution, such temporal variations and the pronounced increase in dust concentration (e.g., Al concentration reached 3.72 μg m$^{-3}$ in Sample 11) would not be expected.

In summary, although these three samples were more affected by anthropogenic pollution compared to most of other samples, the dominant influence was from long-range transport rather than local sources. These continental aerosols, carried by strong northerly winds, can be further transported to downwind oceanic regions. Therefore, we did not treat these samples separately in our study but included a brief discussion of the long-range transport in the revised manuscript.

[Figure]

Figure C2. 72-hour air mass backward trajectories arriving at the sample site for (a) Sample 9 and (b) Samples 10-11 in autumn campaign. The color of each trajectory point represents the corresponding time before arriving at the sampling sites.

**Lines 385-390 (398-405):** The two samples corresponding to event T3 were collected at the same location near the coast. In this event, peaks in anthropogenic pollutant concentrations and PMF source contributions from primary anthropogenic emissions and CS&B appeared in Sample 10, preceding the peak in dust concentration observed in Sample 11 (Fig. 9). This temporal pattern is consistent with the dynamics of long-range transport, where pollutants from continental China typically arrive before dust originating from more distant inland regions. Local anthropogenic emissions in the coastal area played a secondary role in shaping the observed aerosol concentrations and properties during this event.

3.  Line 149: Given the large uncertainties in PM mass concentration, the comparison here does not appear convincing.

Thank you for your insightful comment. We agree that there are considerable uncertainties in PM mass concentration measurements, particularly during the autumn and winter campaigns, likely due to the change in filter membrane type from PTFE to polycarbonate. However, our comparison focuses on campaign-average concentrations. Based on error propagation principles, the uncertainty of a group average ($Unc_{\bar{X}}$) can be estimated as $Unc_{\bar{X}} = \sqrt{\frac{Unc_X}{N}}$, where $Unc_X$ is the uncertainty for an individual measurement and $N$ is the number of samples in the group. Assuming a relative uncertainty of up to 100% for individual measurements in the autumn and winter campaigns ($N = 16$ and 18, respectively), the resulting relative uncertainty for the campaign-average concentrations would be significantly lower ($Unc_{\bar{X}} \leq 25\%$). Therefore, we believe that statistical comparisons based on campaign averages remain valid. We have added a brief discussion of PM mass measurement uncertainties in the revised manuscript.

**Lines 163-165 (172-175):** It is worth noting that although individual PM mass measurements, particularly during the autumn and winter campaigns, are subject to substantial uncertainties, the uncertainties associated with campaign-averaged values are much lower, making seasonal comparisons of the averages still reasonable.

4.  Section 3.2: The results for EC measurements are missing.

Thank you for your comment. The concentration of EC was too low to be accurately quantified in many samples. In particular, during the summer, EC levels on 86% of the sampling filters were below the detection limit. As a result, we did not include EC in the discussion of our study. We have now added this clarification to the *Methodology* section.

**Lines 108-109 (115-116):** However, due to the low EC concentrations and limited detectability (e.g., below the detection limit in 86% of samples during the summer campaign), the EC data were excluded from subsequent analyses.

5.  Line 202: "cme"?

This should be "come". Thank you for pointing out this typo. We have corrected it.

6.  Lines 254–255: Please list the sample numbers for the co-transport events. How were these events identified? Have the authors examined wind data during these events? Does the wind information support the interpretation?

Thank you for your suggestions. In response to your comment and that of Reviewer #1, we have now listed the sampling numbers and corresponding dates for the three co-transport events and marked them in the time series of Figure 9 (previously Figure 8). These events were identified based on the time

series of PMF factor contribution intensities. Specifically, they correspond to concurrent strong increases in the contribution of the dust factor and two continental pollution factors, i.e., primary anthropogenic emissions and CS&B. This interpretation is further supported by air mass backward trajectories. As shown in Figure S9, during all three identified events, a portion of the air masses originated from the Gobi Desert in Mongolia and subsequently passed through densely populated regions of northern and eastern China.

**Lines 379-385 (393-398):** In this study, three strong co-transport events of dust and continental pollution aerosols (27 March – 28 March 2017, Sample 1; 14 April 2017, Sample 8 and 9; 22 October – 24 October 2020, Sample 10 and 11) were observed, as denoted by the events T1 – T3 in Fig. 9. These events are characterized by concurrent strong increases in the contribution intensity of three PMF factors: dust, primary anthropogenic emissions, and CS&B. Backward trajectory analyses (Fig. S9) indicate that the air masses associated with these events originated from the Gobi Desert in Mongolia and passed through the densely populated regions of northern and eastern China.

[Figure]

Figure 9. Temporal variations of the normalized contribution intensity of different sources and the mass concentrations of fine- and coarse-mode aerosols. The x-axis label represents the sample ID in chronological order within each cruise. T1 – T3 with brown shaded bands denote the three co-transport events of dust and continental pollution events.

[Figure]

Figure S9. 72-hour air mass backward trajectories during three co-transport events of dust and continental pollution aerosols: (a) Sample 1 from the spring campaign, (b) Samples 8-9 from the spring campaign, and (c) Samples 10-11 from the autumn campaign. The color of each trajectory point indicates the time before arrival at the sampling sites.

7.  Lines 266–270: The authors discuss coarse-mode NO3 formation via nitrogen-containing gases in the previous paragraph and now discuss a transformation from fine-mode to coarse-mode particles. Do the authors suggest that nitrogen gases typically react with fine particles to form particle-phase NO3 in the absence of dust? The relationship between nitrogen gases, fine-mode NO3, and coarse-mode NO3 needs clarification.

Thank you for your question. In polluted continental environments with elevated levels of atmospheric $NO_x$ and $NH_3$, $NO_3^-$ is commonly present as $NH_4NO_3$ and primarily resides in fine particles. However, $NH_4NO_3$ is thermodynamically unstable and can decompose into $HNO_3$ and $NH_3$ gases under warm temperature conditions or when the ambient concentrations of $HNO_3$ and $NH_3$ are low. In the presence of dust or SSA particles, they can uptake gaseous $HNO_3$, forming more stable nitrates in coarse mode. This uptake reduces the atmospheric concentration of $HNO_3$, thereby shifting the gas-particle equilibrium of $NH_4NO_3$ towards the gas phases and consequently decreasing the concentration of fine-mode $NO_3^-$. In the revised manuscript, we have added a detailed discussion of these processes and the relationships between nitrogen gases, fine-mode $NO_3^-$, and coarse-mode $NO_3^-$ at the beginning of the *Discussion* section.

**Lines 268-288 (281-301):**

In polluted continental environments with abundant $NO_x$ and $NH_3$ in the atmosphere, $NO_3^-$ primarily exists in the form of $NH_4NO_3$ and resides in fine particles (Guo et al., 2010), which is captured by the CS&B factor in our SR-PMF analysis. However, $NH_4NO_3$ is thermodynamically unstable and can decompose into gaseous $HNO_3$ and $NH_3$ under warm temperatures or when the ambient concentrations of $HNO_3$ and $NH_3$ are low (Guo et al., 2010; Uno et al., 2017):

$$NH_4NO_3 \rightleftharpoons HNO_3(g) + NH_3(g) \qquad (1)$$

As continental air masses are transported over the ocean, the concentrations of gaseous $HNO_3$ and $NH_3$ decline rapidly due to the substantially lower emission fluxes of $NO_x$ and $NH_3$ in marine environments. This shifts the gas-particle equilibrium toward the gas phase, resulting in the gradual decomposition of

fine-mode $NH_4NO_3$.

In the presence of mineral dust or SSA particles, gaseous $HNO_3$ can be taken up and undergo heterogeneous reactions with specific components such as $CaCO_3$ and $NaCl$, leading to the formation of thermodynamically stable nitrate compounds like $Ca(NO_3)_2$ and $NaNO_3$ as shown below (Usher et al., 2003; Rossi, 2003; Bondy et al., 2017; Liu et al., 2022a). In addition to $HNO_3$, dust and SSA aerosols can also uptake other nitrogen-containing acidic gases (e.g., $N_2O_5$, $NO_2$), which similarly undergo heterogeneous reactions to form stable nitrates (Rossi, 2003; Tang et al., 2016). For SSA, these reactions also result in the conversion of $Cl^-$ to volatile chlorine-containing gases (e.g., $HCl$, $ClNO_2$), causing sea salt chloride depletion.

$$2HNO_3(g) + CaCO_3 \longrightarrow Ca(NO_3)_2 + CO_2(g) + H_2O \qquad (2)$$

$$HNO_3(g) + NaCl \longrightarrow NaNO_3 + HCl(g) \qquad (3)$$

$$N_2O_5(g) + NaCl \longrightarrow NaNO_3 + ClNO_2(g) \qquad (4)$$

Collectively, these heterogeneous reactions further reduce the atmospheric concentration of gaseous $HNO_3$, thus promoting the decomposition and inhibiting the reformation of fine-mode $NH_4NO_3$. Since dust and SSA particles are mainly distributed in the coarse mode, the resulting nitrate products are likewise primarily found in coarse particles. This leads to a redistribution of $NO_3^-$ from the fine to the coarse mode.

8.  Lines 281–283: Is there any data comparing the formation of NaNO3 dust-associated NO3?

Yes, our PMF analysis result quantified the concentrations of SSA-associated and dust-associated $NO_3^-$, represented by the aged sea salt and dust factors, respectively. The overall contributions are presented in the PMF profiles in Figure 6 and Figure S6 (previously Figure S4), the size-resolved contributions are shown in Figure 12a (previously Figure 9a) and Figure S11 (previously Figure 10). $NaNO_3$ formed in aged SSA accounts for 43.2% of the total $NO_3^-$ concentration, whereas dust-associated $NO_3^-$ contributes 12.7%. We have reorganized the *Discussion* section, and this comparison is discussed in detail in Sections 4.3.1 and 4.5.

9.  Line 285: The meaning of this sentence is unclear. Please revise.

This sentence has been removed during the reorganization of the manuscript. The evolution of continental secondary aerosols and the $NO_3^-$ redistribution processes are now discussed in detail in Section 4.1.1 (**Lines 268-308**).

10. Is there data in this study that supports the proposed transformation mechanism of NO3 SO4?

We have added an analysis of the evolution of the $NO_3^-$ size distribution in relation to air mass transport history, air temperature, and SSA concentration. The results clearly show that the concentration ratio between fine-mode and coarse-mode $NO_3^-$ gradually decreases as air masses spend more time traveling over the ocean (i.e., as continental aerosols become more aged). Additionally, higher air temperatures

and SSA concentrations appear to facilitate this fine-to-coarse transition (Figure 8), supporting the proposed transformation mechanism involving the decomposition of fine-mode $NH_4NO_3$ and heterogeneous reactions between nitrogen-containing acidic gases with coarse-mode SSA (as well as dust aerosols during strong dust transport events).

In contrast, fine-mode $SO_4^{2-}$ is thermodynamically stable and less prone to decomposition, due to the very low saturated vapor pressure of $H_2SO_4$ compared to $HNO_3$. Consistent with this, our data show that $SO_4^{2-}$ remains predominantly in the fine mode even during the summer cruise, when the aging degree of continental aerosols is relatively high (Figure 3c). Furthermore, no significant correlation is observed between the fine-to-coarse ratio of $SO_4^{2-}$ and the air mass travel time over the ocean (Figure C3). These observations suggest that $SO_4^{2-}$ does not undergo a similar transformation.

[Figure]

Figure 8. (a) Scattering plot between the concentration ratio of fine-mode to coarse-mode $NO_3^-$ (Fine-to-coarse ratio$_{[NO_3^-]}$) and the time that 72-hour air mass backward trajectories spent over the ocean (Time over ocean). The color and size of the scatter points indicate the average air temperature along the trajectory (Temperature$_{traj}$) and the size-integrated $Na^+$ concentration, respectively. The two filled circles highlight cases observed during the strongest dust event (22 October – 24 October 2020). (b) Boxcharts of the Fine-to-coarse ratio$_{[NO_3^-]}$ grouped by different ranges of Time over ocean. The boxes represent the interquartile range (25th to 75th percentiles), the horizontal lines indicate the median, and the whiskers represent the highest and lowest values within median ± 1.5 interquartile range. Red circles denote the mean values, and stars indicate statistical outliers.

[Figure]

Figure C3. (a) Scattering plot between concentration ratio of fine-mode to coarse-mode $SO_4^{2-}$ (Fine-to-coarse ratio$_{[SO_4^{2-}]}$) and the time that 72-hour air mass backward trajectories spent over the ocean (Time over ocean). The color and size of the scatter points indicate the average air temperature along the trajectory (Temperature$_{traj}$) and the size-integrated $Na^+$ concentration, respectively. (b) Boxcharts of the Fine-to-coarse ratio$_{[SO_4^{2-}]}$ grouped by different ranges of Time over ocean.

11. Line 287: Delete "Meanwhile."

Thank you for your suggestion. We have deleted it.

12. Line 296: Please specify the precursors of C2O4. Are these precursors different in continental vs. marine environments?

Thank you for your comments. Based on previous studies, atmospheric oxalate is primarily formed through aqueous-phase oxidation of small organic acids and carbonyl compounds, such as glyoxal, glycolaldehyde, methylglyoxal, glyoxylic acid, and pyruvic acid. These compounds typically originate from the photochemical oxidation of biogenic and anthropogenic VOCs (Myriokefalitakis et al., 2011; Carlton et al., 2007; Myriokefalitakis et al., 2008). The precursors do vary between marine and continental environments. In marine environments, typical precursors include isoprene and unsaturated fatty acids emitted from the sea surface (Kawamura and Sakaguchi, 1999; Cui et al., 2023; Boreddy et al., 2017). In contrast, atmospheric oxalate in continental environments are influenced not only by biogenic sources such as isoprene and terpenes from forests, but also by a wide range of anthropogenic VOCs, such as acetylene, ethene, and aromatic compounds (Myriokefalitakis et al., 2008).

However, due to the lack of simultaneous measurements of VOC species in our study, we are unable to explicitly identify the specific precursors or distinguish the relative contributions from marine and continental sources. In response to the comment, we have added a brief discussion of potential precursors based on relevant literature.

**Lines 352-359 (365-372):** Additionally, both continental and marine environments emit precursors of $C_2O_4^{2-}$, such as isoprene and terpenes from the forests, acetylene and aromatics from anthropogenic sources, and isoprene and unsaturated fatty acids from the ocean surface (Myriokefalitakis et al., 2008; Kawamura and Sakaguchi, 1999; Cui et al., 2023; Boreddy et al., 2017; Myriokefalitakis et al., 2011). These precursors can undergo photochemical oxidation to form small organic acids and carbonyl compounds, which are further oxidized to $C_2O_4^{2-}$ via aqueous-phase reactions (Myriokefalitakis et al., 2011; Carlton et al., 2007; Myriokefalitakis et al., 2008). Previous studies have highlighted a mixing contribution of marine and continental origins to $C_2O_4^{2-}$ in coastal regions (Wang et al., 2016; Zhou et al., 2015), which is also consistent with our SR-PMF results.

13. Lines 305–306: Is the unimodal distribution of NO3 more likely due to the lack of fine-mode NH4NO3 from land sources, rather than transformation to coarse particles?

That is a great question. We believe both the lack of direct continental aerosol input from land sources and the transformation to coarse particles contribute to the observed unimodal distribution. The former was already implied in our previous statement: "*In summer, as air masses primarily originate from the southeastern ocean, the direct influence of terrestrial transport and thus the overall contribution by CS&B is significantly weaker*". We have rephased the sentence to clarify this point more explicitly.

**Lines 319-323 (332-336):** In summer, as air masses primarily originate from the southeastern ocean (Fig. S2), the direct influence of terrestrial transport and thus the overall contribution from CS&B is significantly weaker (Fig. 9). Correspondingly, the continental input of fine-mode $NH_4NO_3$ is limited, and any continental aerosols present are likely highly aged due to extended transport times over the ocean. Higher temperatures in summer may further promote the decomposition and aging of fine-mode $NH_4NO_3$.

14.   The authors frequently refer to aerosol aging. Were there any measurements related to chemical composition (e.g., O:C ratio)? Or is the discussion of aging solely based on PMF results?

Thank you for your question. In this study, aerosol aging refers to the $NO_3^-$-related transformation processes affecting three types of aerosols: continental secondary aerosols, dust aerosols, and sea salt aerosols. Unfortunately, we do not have detailed measurements of organic composition or the O:C ratio. For continental secondary aerosols, aging is assessed and discussed primarily based on the size distribution of $NO_3^-$ and the air mass travel time over the ocean, as detailed in Section 4.1.1. For dust aerosols, aging is characterized mainly through PMF analysis. For sea salt aerosols, in addition to PMF results, the extent of chloride depletion is also used as a quantitative indicator of aging.

15.   Figure S5: It appears the R² is driven primarily by the highest x-axis data point.

We agree that the data distributions of the variables used in the regression analyses are not perfectly normal, and that high values may have inflated the $R^2$ to some extent. In response to your comment and Reviewer #1's suggestion, we have revised the relevant figures and updated the discussion in the main text. The interpretation now focuses primarily on the regression slope rather than the R² value. More details can be found in our response to Reviewer #1's Comment 11 (Pages 14-15 of this document).

**Lines 360-369 (373-382):** Summing the contributions of each PMF factor to the total concentrations of the chemical species included in the SR-PMF analysis, we find that the combined contribution from CS&B and AC&MS factors is strongly correlated with the $PM_{1.8}$ mass concentration ($R^2 = 0.678$; Fig. S8a). It is important to note that organics were not included in the SR-PMF analysis due to the lack of measurement during the autumn and winter campaigns. Consequently, the PMF-reconstructed concentrations are significantly lower than the observed $PM_{1.8}$ mass concentrations, with a linear regression slope of 0.366 (Fig. S8a). Including other PMF factors only slightly improves the regression slope, from 0.366 to 0.384 (Fig. S8). Organic aerosols are known to be abundant in fine particles over eastern China, with concentrations comparable to those of inorganic components, and they predominantly originate from combustion-related sources and secondary formation processes (Daellenbach et al., 2024; Huang et al., 2014). Therefore, it is likely that CS&B and AC&MS represent the primary sources of fine particles in our study, especially during spring, autumn, and winter.

[Figure]

Figure S8. Correlations between PM$_{1.8}$ mass concentrations and the total concentrations of PMF species in sub-1.8 μm particles contributed by (a) CS&B and AC&MS, and (b) all PMF factors. Here PMF species correspond to all chemical species included in PMF analysis but excluding duplicate species (e.g., Na$^+$ is excluded because elemental Na is also included). Different colors represent different seasons.

16. Are the authors equating the R² value to "70% variance explained"? If so, this may not be statistically accurate.

Thanks for raising this issue, which was also noted by Reviewer #1. As mentioned in our responses to the previous comment and to Reviewer #1's Comments 11 and 12, we have rephrased the sentence and removed the discussion regarding the proportion of variance explained.

17. Lines 334–335: In the PMF "aged SSA" factor and the related discussion, Cl appears to be zero. Is this how aged SSA is defined in this study or in the literature? Is such a complete depletion realistic in ambient conditions?

Thank you for your great question. In the real atmosphere, SSA particles are typically only partially aged. An SSA particle can be viewed as an internal mixture of unreacted sea salt components and those in which Cl$^-$ has been replaced by NO$_3^-$, which are represented by the fresh sea salt and aged sea salt PMF factors, respectively. In other words, these two PMF factors should not be interpreted as representing an external mixture of fully fresh and fully aged SSA particles. Therefore, the actual aging state of SSA is reflected by the combined contributions of these two factors. We have added a clarification on this point in the revised manuscript.

**Lines 417-422 (492-496):** It is important to note that in the real atmosphere, SSA particles are typically only partially aged. An SSA particle can be viewed as an internal mixture of unreacted sea salt components and those in which Cl$^-$ has been replaced by NO$_3^-$, which are represented by the fresh sea salt and aged sea salt PMF factors, respectively. In other words, these two PMF factors do not represent an external mixture of entirely fresh and entirely aged SSA particles. The combined contributions of these two factors reflect the actual aging state of SSA in the atmosphere.

18. Line 335: The authors use the SO4/Na ratio from seawater to estimate SO4. Does this assume that the ratio remains constant even in aged aerosols?

Here we are estimating the concentration of primary $SO_4^{2-}$ from sea spray emission (sea-salt $SO_4^{2-}$). Although the ratio of total $SO_4^{2-}$ to $Na^+$ can increase during atmospheric aging due to the formation of secondary sulfate on SSA particles, we assume that the ratio of sea-salt $SO_4^{2-}$ to $Na^+$ remains constant. This assumption is based on the chemical stability and non-volatility of sea-salt $SO_4^{2-}$ and widely accepted in literature. Many previous studies have adopted the same approach to distinguish between sea-salt $SO_4^{2-}$ and non-sea-salt $SO_4^{2-}$ (Li et al., 2018; Keene et al., 2007; Jongebloed et al., 2023). In the revised manuscript, we have specified the value of this ratio and included a reference to support it.

**Lines 415-416 (489-490):** After subtracting sea salt-derived $SO_4^{2-}$ based on the seawater $SO_4^{2-}/Na^+$ molar ratio of 0.058 (Keene et al., 2007),

19. Lines 341–342: What data supports this calculation and conclusion?

This is supported by the PMF analysis results. On average, the contributions of the aged sea salt and dust factors to $NO_3^-$ were 1.47 μg m$^{-3}$ and 0.43 μg m$^{-3}$, accounting for 43.2% and 12.7% of the total $NO_3^-$ concentration, respectively. We have added this information and the specific values to the revised manuscript.

**Lines 423-427 (497-503):** As noted in Section 4.1, the heterogeneous reactions between SSA and nitrogen-containing acidic gases shift $NO_3^-$ from smaller to larger particle sizes (Chen et al., 2020), greatly affecting its atmospheric lifetime and transport distance. Based on the SR-PMF results, the average concentration of $NO_3^-$ associated with aged SSA was 1.47 μg m$^{-3}$, accounting for 43.2% of the total $NO_3^-$ concentration (Figs. 6 and S6, Table S3). This contribution is 3.4 times greater than that with dust (i.e., represented by dust factor, 0.43 μg m$^{-3}$), indicating that SSA plays a more significant role in the size redistribution of $NO_3^-$.

20. Lines 343–345: Does "70.7%" refer to $R^2$ and "57.2%" to the slope? Clarify how $R^2$ reflects spatiotemporal variance. This discussion needs expansion.

The value of 57.2% refers to the ratio between the average all-species-integrated concentration from SSA factors and the average PM$_{>1.8}$ concentration, and is not derived from the regression analysis. To avoid potential misinterpretation, we have removed the discussion of the regression results from this section.

**Lines 428-431 (508-514):** Summing the concentration of all species contributed by fresh and aged SSA factors, their combined average concentration accounts for 57.2% of the average PM$_{>1.8}$ mass. This proportion would likely be even higher if components not included in the SR-PMF analysis were considered, indicating SSA generally dominates the coarse-mode aerosol concentration.

21. Lines 351–353: Previously, the authors highlighted that the transformation of NO3 affects its lifetime and deposition. Now, they argue that reactions between SSA and acidic gases occur more

readily on smaller particles. Can the authors compare these two mechanisms and identify which is likely more dominant?

Thank you for your question. The two aspects you mentioned actually refer to different perspectives rather than "two distinct mechanisms". There is only one mechanism involved, i.e., the heterogeneous reaction between nitrogen-containing gases (e.g., $HNO_3$, $NO_2$, $N_2O_5$) and sea salt component (NaCl), in which $Cl^-$ is replaced to form $NaNO_3$. The overall reaction rate is largely controlled by the uptake of these gases onto aerosol surfaces, which depends on aerosol surface area.

Therefore, for SSA particles, smaller fresh particles have a larger specific surface area (i.e., surface area per unit mass) and thus typically experience a higher degree of aging over time. However, due to the size distribution of SSA, both the mass and surface area concentrations are predominantly in the coarse mode (see Figure S10; more detail is provided in our response to Reviewer #1's Comment #17). This suggests that coarse particles contribute more reactive material and surface area overall.

As a result, $NO_3^-$ produced via heterogeneous reactions with SSA is primarily found in the coarse mode. For example, the inorganic sea salt concentration in the 3.2–5.6 μm size range is approximately 50 times higher than in the 0.56–1.0 μm range. Even if the fine-mode sea salt is assumed to be fully aged, and the coarse-mode (3.2–5.6 μm) particles are aged by only 30%, the amount of $NO_3^-$ formed in the coarse mode would still be roughly 15 times greater than in the fine mode.

[Figure]

Figure S10. Normalized size distributions of average concentrations from fresh sea salt and aged sea salt PMF factors, along with the ratio of fresh sea salt to particle diameter ([Fresh sea salt]/$D_p$), which serves as a proxy for the surface area of fresh sea salt. The abnormally high value of [Fresh sea salt]/$D_p$ in the smallest size bin is likely due to large measurement and PMF uncertainties associated with the extremely low sea salt concentrations in that range.

22. Lines 370–371: A reference is needed here.

This is a great suggestion. We have added two references here.

**Lines 467-468 (552-553):** The eastern China seas are nutrient-rich and highly productive, with elevated

concentrations of phytoplankton biomass and dissolved organic matter (He et al., 2013; Hung et al., 2003).

**References:**

He, X., Bai, Y., Pan, D., Chen, C. T. A., Cheng, Q., Wang, D., and Gong, F.: Satellite views of the seasonal and interannual variability of phytoplankton blooms in the eastern China seas over the past 14 yr (1998-2011), Biogeosciences, 10, 4721-4739, 10.5194/bg-10-4721-2013, 2013.

Hung, J. J., Chen, C. H., Gong, G. C., Sheu, D. D., and Shiah, F. K.: Distributions, stoichiometric patterns and cross-shelf exports of dissolved organic matter in the East China Sea, Deep-Sea Res. Pt. II, 50, 1127-1145, 10.1016/s0967-0645(03)00014-6, 2003

23. Lines 395–397: Is there any data regarding reductions in V and Ni concentrations in marine fuels? Given the large drop in aerosol-phase V and Ni in this study, could other factors be involved?

We did not measure V and Ni concentrations in marine fuels in this study. However, previous research has demonstrated reductions in trace metals as a result of improved fuel quality. One example, (i.e., Yu et al. (2021)) has been included and compared with our measurements in the following paragraph. Other studies have similarly observed significant declines in ship-emitted V and Ni concentrations, attributing them to the implementation of marine fuel regulations. At the port of Paranaguá in Brazil, for instance, $PM_{2.5}$-bound V and Ni concentrations decreased by 86.2% and 62.1%, respectively, from 2019 to 2020 following the enforcement of the IMO 2020 regulation (Moreira et al., 2024). Given that shipping emissions are widely recognized as the dominant source of fine-mode V and Ni, it is highly unlikely that such dramatic and universal reductions could be driven by factors other than changes in marine fuel composition.

Although the COVID-19 pandemic may have reduced shipping activity, studies suggest that the overall reduction in Asia was generally within 20% and primarily limited to the first half of 2020 (Yi et al., 2024). Moreover, our winter campaign was conducted before COVID-19 had a significant impact on shipping operations, yet the marked decreases in V and Ni concentrations were already evident. This further supports the conclusion that the pandemic was not the primary driver of the observed reductions.

In the revised manuscript, we have cited relevant references in the sentence: "*A side effect of this regulation is the reduced concentration of certain trace metals in marine fuels*". We also added a brief discussion of the potential impact of COVID-19.

**Lines 497-498 (582-583):** A side effect of this regulation is the reduced concentration of certain trace metals in marine fuels (Yu et al., 2021; Moreira et al., 2024).

**Lines 492-494 (577-579):** The reduction in shipping emission due to the COVID-19 pandemic is not the primary driver for this phenomenon, as the estimated decrease in East Asia was only within 20% (Yi et al., 2024), and the winter campaign was conducted before the pandemic significantly impacted ship activity.

24. Related to comment #4: Since EC is an important marker for shipping emissions and was measured,

the authors should elaborate on EC results and their relevance.

Thank you for your suggestion. As noted in our response to Comment #4, due to the low detectability and substantial uncertainty associated with the EC measurements, we did not elaborate on the EC results in the main discussion.

**Lines 108-109 (115-116):** However, due to the low EC concentrations and limited detectability (e.g., below the detection limit in 86% of samples during the summer campaign), the EC data were excluded from subsequent analyses.

25. Line 448: The section number should be corrected to "5."

Thank you for pointing out this mistake. We have corrected it.

**References**

Bondy, A. L., Wang, B., Laskin, A., Craig, R. L., Nhliziyo, M. V., Bertman, S. B., Pratt, K. A., Shepson, P. B., and Ault, A. P.: Inland Sea Spray Aerosol Transport and Incomplete Chloride Depletion: Varying Degrees of Reactive Processing Observed during SOAS, Environ. Sci. Technol., 51, 9533-9542, 10.1021/acs.est.7b02085, 2017.

Boreddy, S. K. R., Kawamura, K., and Tachibana, E.: Long-term (2001-2013) observations of water-soluble dicarboxylic acids and related compounds over the western North Pacific: trends, seasonality and source apportionment, Scientific Reports, 7, 8518, 10.1038/s41598-017-08745-w, 2017.

Carlton, A. G., Turpin, B. J., Altieri, K. E., Seitzinger, S., Reff, A., Lim, H.-J., and Ervens, B.: Atmospheric oxalic acid and SOA production from glyoxal: Results of aqueous photooxidation experiments, Atmos. Environ., 41, 7588-7602, 10.1016/j.atmosenv.2007.05.035, 2007.

Chen, Y., Cheng, Y., Ma, N., Wei, C., Ran, L., Wolke, R., Größ, J., Wang, Q., Pozzer, A., Denier van der Gon, H. A. C., Spindler, G., Lelieveld, J., Tegen, I., Su, H., and Wiedensohler, A.: Natural sea-salt emissions moderate the climate forcing of anthropogenic nitrate, Atmos. Chem. Phys., 20, 771-786, 10.5194/acp-20-771-2020, 2020.

Cui, L., Xiao, Y., Hu, W., Song, L., Wang, Y., Zhang, C., Fu, P., and Zhu, J.: Enhanced dataset of global marine isoprene emissions from biogenic and photochemical processes for the period 2001–2020, Earth System Science Data, 15, 5403-5425, 10.5194/essd-15-5403-2023, 2023.

Daellenbach, K. R., Cai, J., Hakala, S., Dada, L., Yan, C., Du, W., Yao, L., Zheng, F., Ma, J., Ungeheuer, F., Vogel, A. L., Stolzenburg, D., Hao, Y., Liu, Y., Bianchi, F., Uzu, G., Jaffrezo, J.-L., Worsnop, D. R., Donahue, N. M., and Kulmala, M.: Substantial contribution of transported emissions to organic aerosol in Beijing, Nat. Geosci., 10.1038/s41561-024-01493-3, 2024.

Doumbia, T., Granier, C., Elguindi, N., Bouarar, I., Darras, S., Brasseur, G., Gaubert, B., Liu, Y., Shi, X., Stavrakou, T., Tilmes, S., Lacey, F., Deroubaix, A., and Wang, T.: Changes in global air pollutant emissions during the COVID-19 pandemic: a dataset for atmospheric modeling, Earth System Science Data, 13, 4191-4206, 10.5194/essd-13-4191-2021, 2021.

Fairlie, T. D., Jacob, D. J., Dibb, J. E., Alexander, B., Avery, M. A., van Donkelaar, A., and Zhang, L.: Impact of mineral dust on nitrate, sulfate, and ozone in transpacific Asian pollution plumes, Atmos. Chem. Phys., 10, 3999-4012, 10.5194/acp-10-3999-2010, 2010.

Guo, S., Hu, M., Wang, Z. B., Slanina, J., and Zhao, Y. L.: Size-resolved aerosol water-soluble ionic compositions in the summer of Beijing: implication of regional secondary formation, Atmos. Chem. Phys., 10, 947-959, 10.5194/acp-10-947-2010, 2010.

Huang, R. J., Zhang, Y., Bozzetti, C., Ho, K. F., Cao, J. J., Han, Y., Daellenbach, K. R., Slowik, J. G., Platt, S. M., Canonaco, F., Zotter, P., Wolf, R., Pieber, S. M., Bruns, E. A., Crippa, M., Ciarelli, G., Piazzalunga, A., Schwikowski, M., Abbaszade, G., Schnelle-Kreis, J., Zimmermann, R., An, Z., Szidat, S., Baltensperger, U., El Haddad, I., and Prevot, A. S.: High secondary aerosol contribution to particulate pollution during haze events in China, Nature, 514, 218-222, 10.1038/nature13774, 2014.

Jacobson, M. Z., Turco, R. P., Jensen, E. J., and Toon, O. B.: Modeling coagulation among particles of different composition and size, Atmos. Environ., 28, 1327-1338, 1994.

Jongebloed, U. A., Schauer, A. J., Cole-Dai, J., Larrick, C. G., Porter, W. C., Tashmim, L., Zhai, S., Salimi, S., Edouard, S. R., Geng, L., and Alexander, B.: Industrial-era decline in Arctic methanesulfonic acid is offset by increased biogenic sulfate aerosol, Proc. Natl. Acad. Sci. U. S. A., 120, e2307587120, 10.1073/pnas.2307587120, 2023.

Karydis, V. A., Tsimpidi, A. P., Pozzer, A., Astitha, M., and Lelieveld, J.: Effects of mineral dust on global atmospheric nitrate concentrations, Atmos. Chem. Phys., 16, 1491-1509, 10.5194/acp-16-1491-2016, 2016.

Kawamura, K., and Sakaguchi, F.: Molecular distributions of water soluble dicarboxylic acids in marine aerosols over the Pacific Ocean including tropics, J. Geophys. Res. Atmos., 104, 3501-3509, 10.1029/1998JD100041, 1999.

Keene, W. C., Maring, H., Maben, J. R., Kieber, D. J., Pszenny, A. A. P., Dahl, E. E., Izaguirre, M. A., Davis, A. J., Long, M. S., Zhou, X., Smoydzin, L., and Sander, R.: Chemical and physical characteristics of nascent aerosols produced by bursting bubbles at a model air-sea interface, J. Geophys. Res. Atmos., 112, 10.1029/2007JD008464, 2007.

Li, J., Michalski, G., Davy, P., Harvey, M., Katzman, T., and Wilkins, B.: Investigating Source Contributions of Size-Aggregated Aerosols Collected in Southern Ocean and Baring Head, New Zealand Using Sulfur Isotopes, Geophys. Res. Lett., 45, 3717-3727, 10.1002/2018gl077353, 2018.

Liu, J., Zhang, T., Ding, X., Li, X., Liu, Y., Yan, C., Shen, Y., Yao, X., and Zheng, M.: A clear north-to-south spatial gradiente of chloride in marine aerosol in Chinese seas under the influence of East Asian Winter Monsoon, Sci. Total. Environ., 832, 154929, 10.1016/j.scitotenv.2022.154929, 2022a.

Liu, X., Turner, J. R., Hand, J. L., Schichtel, B. A., and Martin, R. V.: A Global-Scale Mineral Dust Equation, J. Geophys. Res. Atmos., 127, e2022JD036937, 10.1029/2022JD036937, 2022b.

Moreira, C. A. B., Polezer, G., dos Santos Silva, J. C., de Souza Zorzenão, P. C., Godoi, A. F. L., Huergo, L. F., Yamamoto, C. I., de Souza Tadano, Y., Potgieter-Vermaak, S., Reis, R. A., Oliveira, A., and Godoi, R. H. M.: Impact assessment of IMO's sulfur content limits: a case study at latin America's largest grain port, Air Quality, Atmosphere & Health, 17, 2337-2351, 10.1007/s11869-024-01576-5, 2024.

Myhre, G., Grini, A., and Metzger, S.: Modelling of nitrate and ammonium-containing aerosols in presence of sea salt, Atmos. Chem. Phys., 6, 4809-4821, 10.5194/acp-6-4809-2006, 2006.

Myriokefalitakis, S., Vrekoussis, M., Tsigaridis, K., Wittrock, F., Richter, A., Brühl, C., Volkamer, R., Burrows, J. P., and Kanakidou, M.: The influence of natural and anthropogenic secondary sources on the glyoxal global distribution, Atmos. Chem. Phys., 8, 4965-4981, DOI 10.5194/acp-8-4965-2008, 2008.

Myriokefalitakis, S., Tsigaridis, K., Mihalopoulos, N., Sciare, J., Nenes, A., Kawamura, K., Segers, A., and Kanakidou, M.: In-cloud oxalate formation in the global troposphere: a 3-D modeling study, Atmos. Chem. Phys., 11, 5761-5782, 10.5194/acp-11-5761-2011, 2011.

Pakkanen, T. A., Kerminen, V.-M., Hillamo, R. E., Màkinen, M., Màkelà, T., and Virkkula, A.: Distribution of nitrate over sea-salt and soil derived particles—Implications from a field study, J. Atmos. Chem., 24, 189-205, 1996.

Rossi, M. J.: Heterogeneous reactions on salts, Chem. Rev., 103, 4823-4882, 10.1021/cr020507n, 2003.

Savoie, D., and Prospero, J.: Particle size distribution of nitrate and sulfate in the marine atmosphere, Geophys. Res. Lett., 9, 1207-1210, 1982.

Seinfeld, J. H., and Pandis, S. N.: Atmospheric chemistry and physics: from air pollution to climate change, John Wiley & Sons, 2016.

Tang, M., Cziczo, D. J., and Grassian, V. H.: Interactions of Water with Mineral Dust Aerosol: Water Adsorption, Hygroscopicity, Cloud Condensation, and Ice Nucleation, Chem Rev, 116, 4205-4259, 10.1021/acs.chemrev.5b00529, 2016.

Uno, I., Osada, K., Yumimoto, K., Wang, Z., Itahashi, S., Pan, X., Hara, Y., Kanaya, Y., Yamamoto, S., and Fairlie, T. D.: Seasonal variation of fine- and coarse-mode nitrates and related aerosols over East Asia: synergetic observations and chemical transport model analysis, Atmos. Chem. Phys., 17, 14181-14197, 10.5194/acp-17-14181-2017, 2017.

Usher, C. R., Michel, A. E., and Grassian, V. H.: Reactions on mineral dust, Chem Rev, 103, 4883-4940, 10.1021/cr020657y, 2003.

Wu, C., Zhang, S., Wang, G., Lv, S., Li, D., Liu, L., Li, J., Liu, S., Du, W., Meng, J., Qiao, L., Zhou, M., Huang, C., and Wang, H.: Efficient Heterogeneous Formation of Ammonium Nitrate on the Saline Mineral Particle Surface in the Atmosphere of East Asia during Dust Storm Periods, Environ. Sci. Technol., 10.1021/acs.est.0c04544, 2020.

Yi, W., He, T., Wang, X., Soo, Y. H., Luo, Z., Xie, Y., Peng, X., Zhang, W., Wang, Y., Lv, Z., He, K., and Liu, H.: Ship emission variations during the COVID-19 from global and continental perspectives, Sci. Total. Environ., 954, 176633, 10.1016/j.scitotenv.2024.176633, 2024.

Yu, G., Zhang, Y., Yang, F., He, B., Zhang, C., Zou, Z., Yang, X., Li, N., and Chen, J.: Dynamic Ni/V Ratio in the Ship-Emitted Particles Driven by Multiphase Fuel Oil Regulations in Coastal China, Environ. Sci. Technol., 55, 15031-15039, 10.1021/acs.est.1c02612, 2021.

Zheng, B., Zhang, Q., Geng, G., Chen, C., Shi, Q., Cui, M., Lei, Y., and He, K.: Changes in China's anthropogenic emissions and air quality during the COVID-19 pandemic in 2020, Earth System Science Data, 13, 2895-2907, 10.5194/essd-13-2895-2021, 2021.

Zhuang, H., Chan, C. K., Fang, M., and Wexler, A. S.: Formation of nitrate and non-sea-salt sulfate on coarse particles, Atmos. Environ., 33, 4223-4233, 10.1016/s1352-2310(99)00186-7, 1999.